# A Score-Based Model for Learning Neural Wavefunctions

## Abstract

Quantum Monte Carlo coupled with neural network wavefunctions has shown success in computing ground states of quantum many-body systems. Existing optimization approaches compute the energy by sampling local energy from an explicit probability distribution given by the wavefunction. In this work, we provide a new optimization framework for obtaining properties of quantum many-body ground states using score-based neural networks. Our new framework does not require explicit probability distribution and performs the sampling via Langevin dynamics. Our method is based on the key observation that the local energy is directly related to scores, defined as the gradient of the logarithmic wavefunction. Inspired by the score matching and diffusion Monte Carlo methods, we derive a weighted score matching objective to guide our score-based models to converge correctly to ground states. We first evaluate our approach with experiments on quantum harmonic traps, and results show that it can accurately learn ground states of atomic systems. By implicitly modeling high-dimensional data distributions, our work paves the way toward a more efficient representation of quantum systems.

## 1 Introduction

Understanding the properties of quantum systems lies at the core of many scientific disciplines, such as condensed matter physics, material science, and quantum chemistry. A quantum system is characterized by its ground state wavefunction, formally obtained by solving the Schrödinger equation. However, directly solving the Schrödinger equation for quantum systems with many particles is impractical due to the exponentially large Hilbert space. Owning to its strong dimension reduction capabilities, deep learning methods have been used as a strong candidate to approximately solve the Schrödinger equation and extract properties of quantum systems with the desired accuracy. For example, under the supervised learning setting, deep learning methods have been successfully applied to predict the quantum properties of molecular systems based on training data generated from density functional theory (DFT) calculation (Schütt et al., 2017; Gasteiger et al., 2020; Liu et al., 2022; Wang et al., 2022). However, supervised methods rely on expensive computational simulations to generate a large amount of training data, and the accuracy of these methods is fundamentally limited by the data quality. Furthermore, DFT calculations involve various approximations and are not guaranteed to reach true ground states. A common scheme for approximately solving the Schrödinger equation is the variational principle, which optimizes a trial wavefunction to reach the ground state by minimizing its energy as much as possible via quantum Monte Carlo (QMC). Such a method is called variational Monte Carlo, whose accuracy relies on the expressive power of the trial wavefunction. Recently, deep learning methods coupled with variational Monte Carlo have unleashed the potential of both methods (Carleo & Troyer, 2017; Hermann et al., 2022). Powered by the efficient sampling and optimization framework of quantum Monte Carlo and the universal approximation capability of deep neural networks, neural wavefunctions can accurately model quantum states, and dramatic improvements have been achieved (Pfau et al., 2020; Hermann et al., 2020).

Modeling a wavefunction is conceptually similar to modeling a probability density. Existing methods model the wavefunction explicitly by training a neural network to directly output the wavefunction values. However, numerous examples in machine learning have shown that implicitly modeling data distributions provides better representations (Kingma & Welling, 2014; Goodfellow et al., 2014; Ho et al., 2020). As our direct reference, score-based methods have demonstrated their strong suc-

cess in generative modeling (Song & Ermon, 2019; Song et al., 2020). A score is defined as the gradient of the log probability. For example, realistic images can be generated from random noise by following dynamics defined by scores. In this paper, we show that the quantum wavefunction can be represented by score models and be optimized within the QMC framework.

Our motivation to relate score-based models with QMC is based on an interesting connection between energy computations and score-based formulations. In QMC, the energy of a system is averaged over local energy of plausible quantum states. Our observation is that local energy only involves gradients of the logarithmic wavefunction, which we define as the score of the wavefunction. As a result, to minimize energy, the score must be explicitly computed. On the other hand, the actual wavefunction values are only used for sampling and optimization. To this end, we propose a new optimization framework for QMC where sampling and optimization are also achieved by using score functions alone, eliminating the need to explicitly compute the wavefunction value. In our proposed score-based framework, the sampling is done via Langevin dynamics and optimization is done through a new loss function inspired by diffusion Monte Carlo.

Our score-based method enables the possibility of performing QMC computation with only score functions, which is infeasible in existing optimization frameworks. A direct benefit is that, by predicting gradients, we avoid the need to recompute it from the wavefunction. Moreover, score functions can be interpreted as the force of quantum systems, by implicitly modeling distributions with score functions, the dynamics of quantum systems could be better captured. Our experimental results show that with our score-based optimization framework, ground states of quantum systems can be accurately learned.

## 2 BACKGROUND AND RELATED WORK

### 2.1 QUANTUM MANY-BODY WAVEFUNCTION IN CONTINUOUS SPACE

We use $\boldsymbol{x} \in \mathbb{R}^{N \times d}$ to denote the coordinates of $N$ particles in $d$-dimension. The quantum state of a system is defined by its wavefunction $\psi : \mathbb{R}^{N \times d} \to \mathbb{R}$. By definition $\psi$ is normalized ($\int_{\boldsymbol{x}} |\psi(\boldsymbol{x})|^2 = 1$) and $|\psi(\boldsymbol{x})|^2$ gives the probability density of observing $\boldsymbol{x}$. Any wavefunction $\psi$ can be expressed as linear combination of eigenfunctions $\psi_n$, which are solutions to the time-independent Schrödinger equation $\hat{H}\psi_n(\boldsymbol{x}) = E_n\psi_n(\boldsymbol{x})$, where $\hat{H}$ is an linear operator known as the Hamiltonian, and $E_n$ is a scalar giving the energy of the $n$-th eigen state. The Hamiltonian is defined as

$$\hat{H}\psi(\boldsymbol{x}) = -\frac{1}{2}\sum_i \nabla_i^2 \psi(\boldsymbol{x}) + V(\boldsymbol{x})\psi(\boldsymbol{x}), \tag{1}$$

where the index $i$ runs over all of the $N \times d$ dimensions in the summation. The first term in the Hamiltonian takes the sum of the second-order partial derivatives of the wavefunction and is related to the kinetic energy of the system. The second term in the Hamiltonian multiplies the wavefunction by a scalar value and is related to the potential energy of the system. The kinetic term is intrinsic to the Schrödinger equation and always takes the same form, whereas the potential function $V : \mathbb{R}^{N \times d} \to \mathbb{R}$ varies for different physics problems. Note that although a wavefunction can be complex-valued in general, we can let $\psi$ be real-valued because $\hat{H}$ is real.

Our objective is to find the ground state $\psi_0$, which is the eigen state associated with the lowest energy $E_0$. In our notation, the coordinates $\boldsymbol{x}$ can be either viewed as $N$ $d$-dimensional vectors or as a flattened $N \cdot d$ dimensional vector. In the rest of this paper, depending on the context, we may use bold $\boldsymbol{x}_i$ to denote the $i$-th particle or use the regular font $x_i$ to denote the $i$-th scalar component of the flattened vector.

### 2.2 VARIATIONAL MONTE CARLO

The variational Monte Carlo (VMC) method uses a parameterized function $\psi_{\boldsymbol{\theta}} : \mathbb{R}^{N \times d} \to \mathbb{R}$ (called the Ansatz) to model a wavefunction, where $\boldsymbol{\theta}$ denotes the parameters to be optimized. The normalization of $\psi_{\boldsymbol{\theta}}$ is not required. The energy expectation of $\psi_{\boldsymbol{\theta}}$ is computed as:

$$\mathcal{L}(\boldsymbol{\theta}) = \frac{\int \psi_{\boldsymbol{\theta}}(\boldsymbol{x})\hat{H}\psi_{\boldsymbol{\theta}}(\boldsymbol{x})d\boldsymbol{x}}{\int \psi_{\boldsymbol{\theta}}(\boldsymbol{x})\psi_{\boldsymbol{\theta}}(\boldsymbol{x})d\boldsymbol{x}} = \frac{\int \psi_{\boldsymbol{\theta}}(\boldsymbol{x})^2 \frac{\hat{H}\psi_{\boldsymbol{\theta}}(\boldsymbol{x})}{\psi_{\boldsymbol{\theta}}(\boldsymbol{x})}d\boldsymbol{x}}{\int \psi_{\boldsymbol{\theta}}(\boldsymbol{x})^2 d\boldsymbol{x}} = \mathbb{E}_{\boldsymbol{x} \sim \psi_{\boldsymbol{\theta}}^2 / \int \psi_{\boldsymbol{\theta}}^2} E_L(\boldsymbol{x}; \boldsymbol{\theta}), \tag{2}$$

where $E_L(\boldsymbol{x}; \boldsymbol{\theta}) = \frac{\hat{H}\psi_{\boldsymbol{\theta}}(\boldsymbol{x})}{\psi_{\boldsymbol{\theta}}(\boldsymbol{x})}$ is called the *local energy* of $\boldsymbol{x}$. The expectation is evaluated numerically on sparse samples. Markov Chain Monte Carlo (MCMC) sampling is employed to drive sample distribution to converge to the target density $\psi_{\boldsymbol{\theta}}^2 / \int \psi_{\boldsymbol{\theta}}^2$.

We can approach ground state wavefunctions by minimizing the energy expectation with gradient descent. The unbiased gradient of $\mathcal{L}(\boldsymbol{\theta})$ w.r.t. parameters $\boldsymbol{\theta}$ is given by:

$$\nabla_{\boldsymbol{\theta}} \mathcal{L}(\boldsymbol{x}) = 2\mathbb{E}_{\boldsymbol{x} \sim \psi_{\boldsymbol{\theta}}^2 / \int \psi_{\boldsymbol{\theta}}^2} \left[ \left( E_L(\boldsymbol{x}; \boldsymbol{\theta}) - \mathbb{E}_{\boldsymbol{x} \sim \psi_{\boldsymbol{\theta}}^2 / \int \psi_{\boldsymbol{\theta}}^2} [E_L(\boldsymbol{x}; \boldsymbol{\theta})] \right) \nabla_{\boldsymbol{\theta}} \log |\psi_{\boldsymbol{\theta}}(\boldsymbol{x})| \right], \quad (3)$$

where expectations are evaluated by average over samples. The derivation of this loss makes use of the fact that $\hat{H}$ is Hermitian (Ceperley et al., 1977). A detailed derivation can be found in Appendix E of Lin et al. (2021). The optimized energy expectation value gives the approximated ground state energy, and the eigenvalue formulation ensures the estimation to be variational, that is, the energy expectation defined by the Ansatz is always above the true ground state energy. Other properties of quantum systems can also be estimated by taking expectations over corresponding operators.

## 2.3 RELATED WORK

Neural quantum state based on the restrictive Boltzmann machine was initially studied for simple quantum spin models on lattices (Carleo & Troyer, 2017). More complicated neural network architectures, such as convolutional neural network and graph neural network are then extended to more complicated spin systems to capture frustration due to the lattice structure and next nearest neighboring interaction (Kochkov et al., 2021; Fu et al., 2022). Similar ideas are also studied for ab-initio simulation in quantum chemistry (Han et al., 2019) to study properties of small molecules. Recently, FermiNet Pfau et al. (2020) and PauliNet Hermann et al. (2020) greatly improve the accuracy of neural wavefunction and apply to larger molecular systems. Follow-up works contribute to improved performance (Gerard et al., 2022), joint training for multiple geometries (Gao & Günnemann, 2021; Scherbela et al., 2022; Gao & Günnemann, 2022) or solving for excited state (Entwistle et al., 2022). All of the existing methods use neural networks to explicitly model the wavefunction, while in this work we propose to implicitly model the quantum state using the score function.

Diffusion Monte Carlo (DMC) (Toulouse et al., 2016; Ceperley, 2004) employs score-based diffusion to improve upon VMC. In VMC, the quality of approximations is limited by the capacity of Ansatz. DMC additionally assigns a weight for each sample in a way that the weighted distribution gives a better approximation of ground state. In DMC's formulation, each sample is defined by a walker, and each walker is assigned a weight. In importance-sampled DMC, a trial function $\psi_T$ is used. At each iteration, the walkers randomly diffuse by following score of the trial function and the weights are updated according to the imaginary time evolution so that the repeated diffusing and weighting procedure projects out ground states from the trial wavefunction. At convergence, the ground state energy is estimated by the weighted average of local energies. Ceperley & Alder (1980) applies DMC to electonic gas. Ren et al. (2022) and Wilson et al. (2021) apply fixed-node DMC starting from the optimized FermiNet Ansatz.

Score-based methods have shown great success in generative modeling (Song & Ermon, 2019; Song et al., 2020). The effectiveness of implicit density modeling has also been demonstrated by other successful generative models, such as VAEs (Kingma & Welling, 2014), GANs (Goodfellow et al., 2014; Brock et al., 2018), and diffusion models (Sohl-Dickstein et al., 2015; Ho et al., 2020).

## 3 METHOD

For the Hamiltonian $\hat{H}$ defined in Equation 1 the local energy can be expressed in terms of the logarithmic wavefunction as:

$$E_L(\boldsymbol{x}; \boldsymbol{\theta}) = -\frac{1}{2} \sum_i \left( \frac{\partial^2 \log |\psi_{\boldsymbol{\theta}}(\boldsymbol{x})|}{\partial x_i^2} + \left( \frac{\partial \log |\psi_{\boldsymbol{\theta}}(\boldsymbol{x})|}{\partial x_i} \right)^2 \right) + V(\boldsymbol{x}). \quad (4)$$

One can easily prove this expression by using the fact that $\frac{\partial \log |\psi_{\boldsymbol{\theta}}(\boldsymbol{x})|}{\partial x_i} = \frac{1}{x_i} \frac{\partial \psi_{\boldsymbol{\theta}}(\boldsymbol{x})}{\partial x_i}$. If we take a closer look at this expression, we can notice that the local energy depends only on the gradient of

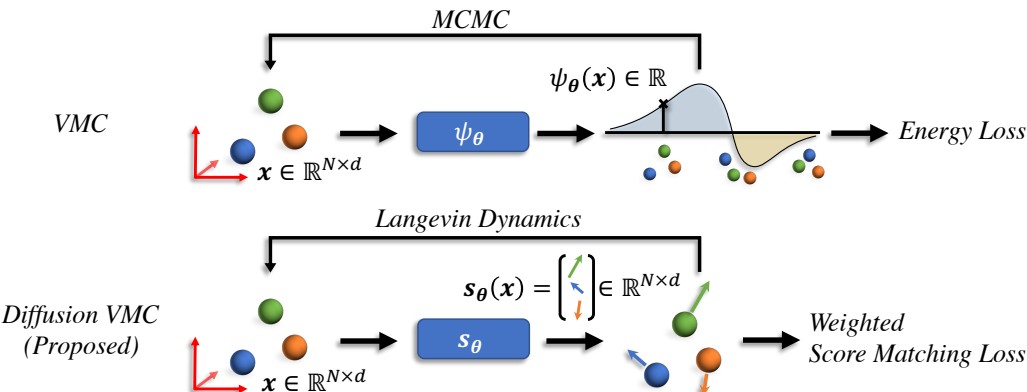

Figure 1: Comparison between the pipeline of VMC and our proposed DiffVMC. In VMC, the wavefunction is explicitly modeled with a neural network, who directly outputs the wavefunction value. The sampling is carried out with MCMC and the optimization is achieved by minimizing the energy loss. In the proposed DiffVMC we instead model the score of the wavefunction. The samples are generated via Langevin dynamics and the proposed weighted score matching objective is employed to optimize the score neural network.

the logarithmic wavefunction. In fact, if we define the function $s_{\boldsymbol{\theta}} : \mathbb{R}^{N \times d} \to \mathbb{R}^{N \times d}$ such that $s_{\boldsymbol{\theta}}(\boldsymbol{x})_i = \frac{\partial \log |\psi_{\boldsymbol{\theta}}(\boldsymbol{x})|}{\partial x_i}$, we can rewrite the local energy as:

$$E_L(\boldsymbol{x}; \boldsymbol{\theta}) = -\frac{1}{2} \left( \text{tr}(\nabla_{\boldsymbol{x}} s_{\boldsymbol{\theta}}(\boldsymbol{x})) + \|s_{\boldsymbol{\theta}}(\boldsymbol{x})\|^2 \right) + V(\boldsymbol{x}). \tag{5}$$

The function $s_{\boldsymbol{\theta}}$ is called the *quantum force* or the *drift* in quantum Monte Carlo. Here we follow the convention from the machine learning community and call it the *score*. This direct connection between local energies and scores motivates us to ask the question: can we represent quantum states using only $s_{\boldsymbol{\theta}}$? As a direct benefit, this can avoid recomputing the first-order derivatives in evaluating the local energy and consequently provide a more succinct representation. It is true that we will lose access to the unnormalized wavefunction. However, in practice, our primary goal is to estimate observables, such as kinetic energy or density, which can be estimated from sample distributions.

To this end, our score-based optimization framework follows the gradient descent formulation in VMC, and we design a new loss function inspired by DMC. As we mix the flavors of these two methods, we call our new optimization framework the *Diffusion Variational Monte Carlo* (DiffVMC). The pipelines of VMC and DiffVMC are shown in Figure 1.

Please note that the original score (Hyvärinen & Dayan, 2005) is defined in terms of probability densities, so we have $s_{\text{original}}(\boldsymbol{x}) = \nabla_{\boldsymbol{x}} \log p(\boldsymbol{x}) = \nabla_{\boldsymbol{x}} \log \psi(\boldsymbol{x})^2 = 2\nabla_{\boldsymbol{x}} \log |\psi(\boldsymbol{x})|$, which is twice of the score in our definition. By abuse of notation we define our score in terms of wavefunctions. We do this for two reasons. First, wavefunctions are more natural to deal with in quantum mechanics. Second, as we will show later, in addition to moving samples locally, we also use $s_{\boldsymbol{\theta}}(\boldsymbol{x})$ to describe distributions, which shares the same physical meaning as the original score definition.

### 3.1 SCORE-BASED NEURAL WAVEFUNCTION ANSATZ

We use a parameterized score function to implicitly model the wavefunction. Formally, for systems with $N$ particles in $d$ dimension, its score function $s_{\boldsymbol{\theta}} : \mathbb{R}^{N \times d} \to \mathbb{R}^{N \times d}$ maps a set of input coordinates to a set of output scores, which are vectors having the same dimensions as inputs. Intuitively, after training, the output score should tell the particles the direction toward regions with higher density.

In our case, we are dealing with indistinguishable particles in quantum mechanics. So exchanging two particles will not change the probability density: $\psi(\dots, \boldsymbol{x}_i, \dots, \boldsymbol{x}_j, \dots)^2 = \psi(\dots, \boldsymbol{x}_j, \dots, \boldsymbol{x}_i, \dots)^2$. As a result, the score, which is the gradient of a logarithmic wavefunction,

is permutation equivariant. Formally,

$$\nabla_{\boldsymbol{x}_i} \log |\psi(\ldots, \boldsymbol{x}_i, \ldots, \boldsymbol{x}_j, \ldots)| = \nabla_{\boldsymbol{x}_j} \log |\psi(\ldots, \boldsymbol{x}_j, \ldots, \boldsymbol{x}_i, \ldots)|. \qquad (6)$$

Essentially the equivariance means that if two input particles exchange their positions, their corresponding output score vectors will also exchange positions. The proof is straightforward (Appendix B). We parameterize the score function with a neural network. The equivariance can be easily achieved by considering the input coordinates as a set (Zaheer et al., 2017; Qi et al., 2017).

## 3.2 SAMPLING VIA LANGEVIN DYNAMICS

To generate samples that follow the distribution implicitly defined by the score, we use Langevin dynamics, which is similar in the score-based generative models (Song & Ermon, 2019). Given the samples at the current time step $x_t$, the coordinates for the new samples are computed as:

$$\boldsymbol{x}_{t+1} = \boldsymbol{x}_t + \sqrt{\alpha}\boldsymbol{\epsilon} + \alpha \boldsymbol{s}_{\boldsymbol{\theta}}(\boldsymbol{x}_t), \qquad (7)$$

where $\alpha$ is an hyperparameter defining the step size, and $\boldsymbol{\epsilon} \sim \mathcal{N}(\boldsymbol{0}, \boldsymbol{I}^{Nd})$ is a random vector sampled from the standard multivariate Gaussian distribution. The process is the same as in DMC and can be understood as first doing a random diffusion, then drifting by following scores. We can prove that when $\alpha$ is small, the distribution converges to the distribution defined by the score function.

In Langevin dynamics, similar to the Metropolis-Hasting rejection step in MCMC, an accept/reject procedure can be employed to alleviate the finite time error. Although omitted in the context of generative modeling Song & Ermon (2019), this is a standard step in DMC. The original rejection step computes the ratio

$$P_{\mathrm{acc}} = \frac{\exp\left(-\frac{1}{2\alpha}\|\boldsymbol{x} - \boldsymbol{x}' - \boldsymbol{s}_{\boldsymbol{\theta}}(\boldsymbol{x}')\alpha)\|^2\right)\psi_{\boldsymbol{\theta}}(\boldsymbol{x}')^2}{\exp\left(-\frac{1}{2\alpha}\|\boldsymbol{x}' - \boldsymbol{x} - \boldsymbol{s}_{\boldsymbol{\theta}}(\boldsymbol{x})\alpha\|^2\right)\psi_{\boldsymbol{\theta}}(\boldsymbol{x})^2}. \qquad (8)$$

After each Langevin dynamics move, we decide whether to accept or reject the move based on the $P_{\mathrm{acc}}$. Concretly, we first sample a random number uniformly between 0 and 1. The move is accepted if $P_{\mathrm{acc}}$ is larger than the random number and is rejected otherwise. However, the expression of $P_{\mathrm{acc}}$ involves the wavefunction values $\psi_{\boldsymbol{\theta}}(\boldsymbol{x})$ and $\psi_{\boldsymbol{\theta}}(\boldsymbol{x}')$, which are generally not available in our score-based framework. To still be able to use the rejection step, we propose an approximated estimation of $P_{\mathrm{acc}}$ which involves only the score function. This is achieved by approximating $\log \frac{|\psi_{\boldsymbol{\theta}}(\boldsymbol{x}')|}{|\psi_{\boldsymbol{\theta}}(\boldsymbol{x})|} = \log |\psi_{\boldsymbol{\theta}}(\boldsymbol{x}')| - \log |\psi_{\boldsymbol{\theta}}(\boldsymbol{x})|$ using the average gradient $\frac{1}{2}(\nabla_{\boldsymbol{x}} \log |\psi_{\boldsymbol{\theta}}(\boldsymbol{x})| + \nabla_{\boldsymbol{x}} \log |\psi_{\boldsymbol{\theta}}(\boldsymbol{x}'))|$, then we can show that the ratio can be approximated only in terms of the score as (derivation in Appendix C):

$$P_{\mathrm{acc}} \approx \exp\left(\frac{\alpha}{2}(\|\boldsymbol{s}_{\boldsymbol{\theta}}(\boldsymbol{x})\|^2 - \|\boldsymbol{s}_{\boldsymbol{\theta}}(\boldsymbol{x}')\|^2)\right). \qquad (9)$$

## 3.3 NEURAL WAVEFUNCTION OPTIMIZATION

The energy loss in VMC (Equation 3) depends explicitly on wavefunctions. The expression of the energy loss only in terms of scores is unknown. So we need to find a new loss to optimize scores towards ground states. We motivate our new loss from the imaginary time evolution in DMC.

The imaginary time evolution operator $e^{-\tau \hat{H}}$ projects out ground states when $\tau \to \infty$. At short $\tau$, by Taylor expansion, $e^{-\tau \hat{H}}\psi(\boldsymbol{x}) \approx \psi(\boldsymbol{x}) - \tau \hat{H}\psi(\boldsymbol{x}) = \psi(\boldsymbol{x})(1 - \tau \frac{\hat{H}\psi(\boldsymbol{x})}{\psi(\boldsymbol{x})}) \approx e^{-\tau E_L(\boldsymbol{x})}\psi(\boldsymbol{x})$. Thus, evolving $\psi(\boldsymbol{x})$ in imaginary time for a short time $\tau$ can be approximated as:

$$\psi(\boldsymbol{x}) \mapsto \psi'(\boldsymbol{x}) = e^{-\tau E_L(\boldsymbol{x})}\psi(\boldsymbol{x}). \qquad (10)$$

$\psi'$ is closer to the ground state than $\psi$ because higher energy eigenstates decays exponentially faster than the ground state. Therefore, for our score-based model, we can minimize energies by letting $\boldsymbol{s}_{\boldsymbol{\theta}}(\boldsymbol{x})$ approach the evolved score $\nabla_{\boldsymbol{x}} \log |\psi'(\boldsymbol{x})|$. We achieve this via score matching.

Score matching (Hyvärinen & Dayan, 2005) provides a way to make $\boldsymbol{s}_{\boldsymbol{\theta}}$ converge to the true score of the sample distribution. Assume at current step the samples follow $\psi^2 / \int \psi^2$ and $\boldsymbol{s}_{\boldsymbol{\theta}}(\boldsymbol{x}) = \nabla_{\boldsymbol{x}} \log |\psi(\boldsymbol{x})|$, we can make $\boldsymbol{s}_{\boldsymbol{\theta}}(\boldsymbol{x})$ converge to $\nabla_{\boldsymbol{x}} \log |\psi'(\boldsymbol{x})|$ by minimizing the implicit score matching (ISM) objective:

$$\mathrm{ISM}(\boldsymbol{\theta}) = 2\mathbb{E}_{\boldsymbol{x} \sim \psi'^2 / \int \psi'^2}\left[\mathrm{tr}(\nabla_{\boldsymbol{x}} \boldsymbol{s}_{\boldsymbol{\theta}}(\boldsymbol{x})) + \|\boldsymbol{s}_{\boldsymbol{\theta}}(\boldsymbol{x})\|^2\right]. \qquad (11)$$

The problem is that we do not have samples following $\psi'^2 / \int \psi'^2$. We can solve this by transforming the ISM into a weighted version based on current samples:

$$\text{ISM}(\boldsymbol{\theta}) = \frac{2 \int \psi'^2(\boldsymbol{x}) \left[\text{tr}(\nabla_{\boldsymbol{x}} \boldsymbol{s}_{\boldsymbol{\theta}}(\boldsymbol{x})) + \|\boldsymbol{s}_{\boldsymbol{\theta}}(\boldsymbol{x})\|^2\right] d\boldsymbol{x}}{\int \psi'^2(\boldsymbol{x}) d\boldsymbol{x}} \tag{12}$$

$$= \frac{2 \int \psi^2(\boldsymbol{x}) e^{-2\tau E_L(\boldsymbol{x})} \left[\text{tr}(\nabla_{\boldsymbol{x}} \boldsymbol{s}_{\boldsymbol{\theta}}(\boldsymbol{x})) + \|\boldsymbol{s}_{\boldsymbol{\theta}}(\boldsymbol{x})\|^2\right] d\boldsymbol{x}}{\int \psi^2(\boldsymbol{x}) d\boldsymbol{x}} \frac{\int \psi^2(\boldsymbol{x}) d\boldsymbol{x}}{\int \psi'^2(\boldsymbol{x}) d\boldsymbol{x}} \tag{13}$$

$$= 2\mathbb{E}_{\boldsymbol{x} \sim \psi^2 / \int \psi^2} \exp(-2\tau E_L(\boldsymbol{x})) \left[\text{tr}(\nabla_{\boldsymbol{x}} \boldsymbol{s}_{\boldsymbol{\theta}}(\boldsymbol{x})) + \|\boldsymbol{s}_{\boldsymbol{\theta}}(\boldsymbol{x})\|^2\right] d\boldsymbol{x} \cdot C, \tag{14}$$

where $C = \frac{\int \psi^2(\boldsymbol{x}) d\boldsymbol{x}}{\int \psi'^2(\boldsymbol{x}) d\boldsymbol{x}} = \frac{\int \psi(\boldsymbol{x})^2 d\boldsymbol{x}}{\int \exp(-2\tau E_L(\boldsymbol{x})) \psi(\boldsymbol{x})^2} = \left(\mathbb{E}_{\boldsymbol{x} \sim \psi^2 / \int \psi^2} \exp(-2\tau E_L(\boldsymbol{x}))\right)^{-1}$ is a constant independent of $\boldsymbol{\theta}$. In practice, using small $\tau$ makes the loss too small, we thus replace $2\tau$ with a hyperparameter $\beta$. We also subtract the sample mean of local energies to improve numerical stability. Put everything together, given a batch of $M$ samples, we define the weighted score matching (WSM) objective as:

$$E_L(\boldsymbol{x}) = -\frac{1}{2} \left(\text{tr}(\nabla_{\boldsymbol{x}} \boldsymbol{s}_{\boldsymbol{\theta}}(\boldsymbol{x})) + \|\boldsymbol{s}_{\boldsymbol{\theta}}(\boldsymbol{x})\|^2\right) + V(\boldsymbol{x}) \tag{15}$$

$$E_{\text{diff}}(\boldsymbol{x}_i) = E_L(\boldsymbol{x}_i) - \langle E_L \rangle = E_L(\boldsymbol{x}_i) - \frac{1}{M} \sum_{i=1}^{M} E_L(\boldsymbol{x}_i) \tag{16}$$

$$\text{WSM}(\boldsymbol{\theta}) = 2 \sum_{i=1}^{M} \frac{\exp\left(-\beta E_{\text{diff}}(\boldsymbol{x}_i)\right)}{\sum_{i=1}^{M} \exp\left(-\beta E_{\text{diff}}(\boldsymbol{x}_i)\right)} \left[\text{tr}(\nabla_{\boldsymbol{x}} \boldsymbol{s}_{\boldsymbol{\theta}}(\boldsymbol{x}_i)) + \|\boldsymbol{s}_{\boldsymbol{\theta}}(\boldsymbol{x}_i)\|^2\right] \tag{17}$$

$$= 2 \sum_{i=1}^{M} \underbrace{\text{softmax}\left(-\beta E_{\text{diff}}(\boldsymbol{x}_i)\right)_i}_{\text{Does not differentiate w.r.t. } \boldsymbol{\theta}} \underbrace{\left[\text{tr}(\nabla_{\boldsymbol{x}} \boldsymbol{s}_{\boldsymbol{\theta}}(\boldsymbol{x}_i)) + \|\boldsymbol{s}_{\boldsymbol{\theta}}(\boldsymbol{x}_i)\|^2\right]}_{\text{Differentiate w.r.t. } \boldsymbol{\theta}}, \tag{18}$$

where $\text{softmax}(\boldsymbol{a})_i = \frac{\exp(a_i)}{\sum_i \exp(a_i)}$. Note that the weighting terms are treated as constant when differentiating the loss w.r.t. the parameters $\boldsymbol{\theta}$. Only the ISM terms require computing gradient during back-propagation. This weighting scheme is similar to the attention mechanism in machine learning where the attention scores are based on the local energies and $\beta$ plays a similar role to the temperature variable in the Gumbel softmax (Jang et al., 2016). As a interesting observation, the ISM (Equation 11) is -4 times the kinetic part in local energy. So we only need to compute one of them and reuse the computation.

We can prove that WSM is unbiased by using the zero-variance property of ground states. At ground states we have $\hat{\boldsymbol{H}} \psi_0(\boldsymbol{x}) = E_0 \psi_0(\boldsymbol{x})$. Hence $E_L(\boldsymbol{x}) = \frac{\hat{\boldsymbol{H}} \psi_0(\boldsymbol{x})}{\psi_0(\boldsymbol{x})} = E_0$. Consequently, the WSM weights will be identical at ground states and equal to 1 due to softmax. As a result, the WSM loss reduces to the regular ISM loss at the ground state. Moreover, thanks to Langevin dynamics, every state is a local minimum of the ISM loss because the estimated score is the same with the true score of the sample distribution. Hence the ground state is a local minimum of the WSM loss. $\qquad\square$

Also due to the zero-variance property, when close to ground states, the weights should be close to 1. We can create more imbalanced weights by further normalizing $E_{\text{diff}}$ by dividing its standard deviation. We thus have the Scaled-WSM defined as:

$$\text{Scaled-WSM}(\boldsymbol{\theta}) = 2 \sum_{i=1}^{M} \text{softmax}\left(-\beta \frac{E_{\text{diff}}(\boldsymbol{x}_i)}{\text{std}\left(E_{\text{diff}}\right)}\right)_i \left[\text{tr}(\nabla_{\boldsymbol{x}} \boldsymbol{s}_{\boldsymbol{\theta}}(\boldsymbol{x}_i)) + \|\boldsymbol{s}_{\boldsymbol{\theta}}(\boldsymbol{x}_i)\|^2\right]. \tag{19}$$

As in VMC, we update $\boldsymbol{\theta}$ according to the loss for one step, then we run Langevin dynamics for several steps to make samples follow the updated score. In our experiments, we use $\beta \sim 1$. So the short time approximation from imaginary time evolution does not hold strictly. However, since each time we only update parameters for a small step size and that the target wavefunction $\psi'$ is constantly updated, the approximation may still be valid to some extend. But this should be more as an intuition than as an proof. To have further intuition for the convergence behavior, we can compare the WSM loss to the energy gradient 3. In fact, in both cases, we try to increase the probability density for regions with smaller local energies and decrease the probability density for regions with

larger local energies. However, by far we do not have a rigorous proof for the convergence property. Nevertheless, in our experiments, systems converge to ground states consistently with the proposed WSM loss.

### 3.4 The Diffusion Variational Monte Carlo Algorithm

The overall procedure is similar to VMC. We first perform several steps of Langevin dynamics to equilibrate the sampling. Then we compute the weighted score matching loss and update the network parameters via gradient descent. The algorithm is summarized in Algorithm 1.

---

**Algorithm 1** Diffusion Variational Monte Carlo (DiffVMC)

---

**Input:** Randomly initialized sample $\boldsymbol{x}$, score network $\boldsymbol{s_\theta}$, step size $\alpha$, number of iterations $T$, number of Langevin dynamics step $N_{\text{ld}}$
**Output:** New sample $\boldsymbol{x}$, optimized score network $\boldsymbol{s_\theta}$
  **for** $t = 1$ to $T$ **do**
    **for** $i = 1$ to $N_{\text{ld}}$ **do**
      $\boldsymbol{x}' = \boldsymbol{x} + \sqrt{\alpha}\boldsymbol{\epsilon} + \alpha\boldsymbol{s_\theta}(\boldsymbol{x}), \boldsymbol{\epsilon} \sim \mathcal{N}(\boldsymbol{0}, \boldsymbol{I})$        ▷ Langevin dynamics (Section 3.2)
      $P_{\text{acc}} = \exp\left(\frac{\alpha}{2}(\|\boldsymbol{s_\theta}(\boldsymbol{x})\|^2 - \|\boldsymbol{s_\theta}(\boldsymbol{x}')\|^2)\right)$
      Sample $z \sim U(0, 1)$
      **if** $P_{\text{acc}} > z$ **then**
        $\boldsymbol{x} = \boldsymbol{x}'$        ▷ Accept the move
      **end if**
    **end for**
    loss = Scaled-WSM($\boldsymbol{\theta}$)        ▷ Weighted score matching objective (Section 3.3)
    Update $\boldsymbol{\theta}$ via gradient descent to minimize the loss.
  **end for**

---

## 4 Experiments

We first show that our score-based method correctly finds the ground state by solving the harmonic trap. Then we showcase the applicability of our method on simple atomic systems, i.e., interacting fermions with coulomb potential. We will release our code after the review process.

### 4.1 Wavefunction Ansatz for Bosons and Fermions

Since the unnormalized density $\psi^2(\cdot)$ is invariant under exchange of particle positions (Section 3.1), there are two cases for wavefunction values. The first case is that the wavefunction does not change its sign, i.e., $\psi(\ldots, \boldsymbol{x}_i, \ldots, \boldsymbol{x}_j, \ldots) = \psi(\ldots, \boldsymbol{x}_j, \ldots, \boldsymbol{x}_i, \ldots)$, such particles are called *bosons*. The second case is that the wavefunction change its sign, i.e., $\psi(\ldots, \boldsymbol{x}_i, \ldots, \boldsymbol{x}_j, \ldots) = -\psi(\ldots, \boldsymbol{x}_j, \ldots, \boldsymbol{x}_i, \ldots)$, such particles are called *fermions*. These different symmetries will fundamentally change ground states. The fermion ground state must consider the antisymmetric constraint and has higher ground state energy than the boson ground state.

**Bosons.** We use a feed-forward neural network (Figure 3a) composed of multi-layer perceptrons (MLPs) which satisfies the permutation equivariance (Zaheer et al., 2017; Qi et al., 2017). Input features are coordinates and distances to the center. Each particle is simultaneously transformed into two feature vectors with two different MLPs. We perform average pooling for the first feature vector over all particles and obtain a global feature. The global feature is then concatenated with the second feature vector. A final MLP is used to predict the score of each particle. All MLPs are shared among different particles.

**Fermions.** The antisymmetric property of fermions is very challenging to model (Ceperley, 1991). Different from bosons, due to the change of sign, the fermion wavefunction has both positive regions and negative regions. The region where the wavefunction equals zero is called the nodal surface (or the node). In 1-d, the nodal surface is exactly the set where two particles coincide. However, for higher dimension the nodal surface can be arbitrary. The fermion sign structure gives rise some intrinsic difficulties to parametrize the score function.The score function diverges inversely proportionally to the distance away from the nodal surface (Umrigar et al., 1993). Since neural networks

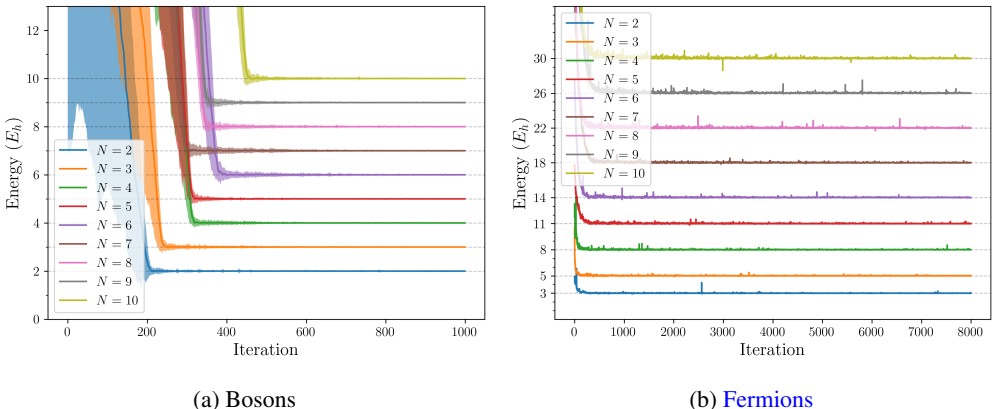

(a) Bosons
(b) Fermions

Figure 2: Train energies for bosons and fermions in 2d quantum harmonic trap. $N$ is the number of particles. All runs converge correctly to ground states, despite slight fluctuations for fermions.

essentially models continuous functions, using feed-forward neural networks to model the score function is prohibitive. In fact, even the networks manage to approximately model the discontinuity, in order to compute the local energy we also requires the derivatives of the score to be accurate, which is more difficult.

Nevertheless, we are still able to illustrate how our score-based framework works with fermions. We can do so by modeling the score as the gradient of an antisymmetric wavefunction Ansatz. We use the score computed from FermiNet (Pfau et al., 2020), where Slater determinants are employed to ensure the antisymmetry. We call it $\nabla_{\boldsymbol{x}}$FermiNet (Figure 3b). Although by doing so the wavefunction value is practically computed, our goal is to evaluate the our method in this more challenging fermion setting. Currently this stands as the only feasible option to overcome fermion sign problem. More generally we can model scores for fermions as the sum of the gradient of an antisymmetric wavefunction and a symmetric wavefunction:

$$\boldsymbol{s}_{\text{Fermion}}(\boldsymbol{x}) = \nabla_{\boldsymbol{x}} f_{\boldsymbol{\theta}_1}(\boldsymbol{x}) + \boldsymbol{g}_{\boldsymbol{\theta}_2}(\boldsymbol{x}) - \boldsymbol{x}. \tag{20}$$

In our setting, $f_{\boldsymbol{\theta}_1}$ is FermiNet and we can model $\boldsymbol{g}_{\boldsymbol{\theta}_2}$ using a feed-forward network (FFN). In our experiment, FFN uses FermiNet encoder to get a feature vector for each particle, which is then mapped to score via a linear layer. The output of FermiNet score and FFN score are summed together to get the final score. We call it $\nabla_{\boldsymbol{x}}$FermiNet+FFN (Figure 3c). We do $-\boldsymbol{x}$ to ensure density vanishes at boundary.

## 4.2 2D QUANTUM HARMONIC TRAP

The Schrödinger equation with harmonic potential

$$V_{\text{QHO}}(\boldsymbol{x}) = \frac{1}{2}\|\boldsymbol{x}\|^2. \tag{21}$$

describes a quantum harmonic oscillator, one of the most famous quantum systems that is solvable. The energy levels and eigenstates are known analytically. In particular, for $n$ bosons in the $d$ dimensional harmonic potential, the ground state energy and wavefunction are $E = \frac{nd}{2}$, $|\psi\rangle = \frac{1}{2^{nd/2}}\exp(-\frac{1}{2}\|\boldsymbol{x}\|^2)$. Therefore, the quantum harmonic oscillator provides an analytical benchmark for our numerical method. Furthermore, despite its simplicity, the quantum harmonic oscillator is of important experimental relevance and can be realized in cold atom systems by trapping atoms using lasers (Dalfovo et al., 1999). Ground states for fermions are more complex due to the antisymmetry constraint. The ground state energies are $E_n = \sum_{i=1}^{n} e_i$ where $e = 1, 2, 2, 3, 3, 3, 4 \ldots$ and the ground state wavefunctions are Slater determinants of Hermite polynomials. $\nabla_{\boldsymbol{x}}$FermiNet+FFN is used here. We conduct simulation with various number of particles. The results are shown in Figure 2. All runs converge correctly to ground states.

### 4.3 Atomic Systems

For atoms, we work in the Born-Oppenheimer where we assume the atoms are fixed in space and only the electrons are allowed to move. Hence, our inputs are the coordinates of the electrons. The potential due to the Coulomb interactions is:

$$V_{\text{Atom}}(\boldsymbol{x}) = \sum_{i>j} \frac{1}{\|\boldsymbol{x}_i - \boldsymbol{x}_j\|} - \sum_i \frac{Z}{\|\boldsymbol{x}_i\|}, \tag{22}$$

where $Z$ is the atom charge. As mentioned above, we compute the score by taking the gradient of FermiNet (Pfau et al., 2020). Following Lin et al. (2021), four atoms are tested, including Boron, Carbon, Nitrogen and Oxygen. To show the effect of step size, We trained and tested the score network with $\alpha$=1e-3 and $\alpha$=1e-2. We and optimize for 200k iterations for all systems. We use $\beta$=2 for Oxygen and $\beta$=1 for all other atoms. We do Langevin dynamics for 20 steps between two parameter update. The norm of the score is clipped at 20 to increase numerical stability. The complete hyperparameters are in Appendix D.

| | $\nabla_{\boldsymbol{x}}$FermiNet $\alpha$=1e-3 | $\nabla_{\boldsymbol{x}}$FermiNet $\alpha$=1e-2 | $\nabla_{\boldsymbol{x}}$FermiNet+FFN $\alpha$=1e-3 | FermiNet | HF | Reference |
|---|---|---|---|---|---|---|
| B | -24.64893(22) | -24.65233(3) | -24.63537(46) | -24.65370(3) | -24.53316 | -24.65391 |
| C | -37.84414(13) | -37.83445(5) | -37.83736(43) | -37.84471(5) | -37.6938 | -37.8450 |
| N | -54.58171(29) | -54.53580(12) | -54.58221(28) | -54.58882(6) | -54.4047 | -54.5892 |
| O | -75.06587(63) | -74.94880(709) | -75.06410(53) | -75.06655(7) | -74.8192 | -75.0673 |

Table 1: Experimental results on Atoms in Hartree ($E_h$). Underline denotes our results within chemical accuracy. References taken from Lin et al. (2021).

The results are summarized on Table 1. The chemical accuracy is defined as 1.594 m$E_h$ (Pfau et al., 2020). Except for the Nitrogen atom, all atoms can enter the chemical accuracy under one setting, although the effect of step size is mixed. In many cases a higher level of variance is observed. The reason may be due to the difficulty in optimizing the gradient network. Moreover, FermiNet works with the second order optimizer KFAC (Martens & Grosse, 2015), which is adapted for wavefunction outputs and may not be suitable for our socre-based optimization. Nevertheless, our framework has demonstrated the correct convergence behaviour for the challenging electronic potential. The results for N are improved with FFN. For other atoms the results are not improved but are comparable. With this setting we show that in terms of network architectures we can go beyond the FermiNet score.

## 5 Comparison to VMC and DMC

Compared to DMC, DiffVMC is closer to VMC's paradigm where Ansatz is optimized by estimating loss based on samples following Ansatz distribution. With DiffVMC we are able to directly model scores, which are more fundamental to the optimization problem and could be more expressive than modeling wavefunctions. On the other hand, DiffVMC and DMC both use scores to update samples. DMC starts with an optimized trial wavefunction Ansatz. Energy minimization is achieved by constantly adjusting weights of walkers. However, the walkers are always guided by the trial function's score and the parameters of the trial Ansatz cannot be updated in DMC. In contrast, DiffVMC is able to update the guiding score function directly. Nevertheless, there is no conflict between DiffVMC and DMC. DMC walkers can be guided with a score optimized with DiffVMC and further project toward ground state through weighting.

## 6 Conclusion

Inspired by the connection between the score-based formulation of local energy, we explore the possibility to implicitly model the quantum wavefunction. With the weighted score matching objective, the proposed DiffVMC enables the possibility to optimize the score network toward the ground state. Experiments show that our proposed method can accurately find the ground state for both bosons and fermions.

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

## A  COMPUTATIONAL COMPLEXITY

Compared to VMC, DiffVMC avoids recomputing gradient of wavefunctions during energy evaluation, but the loss is more complicated. So the overall computational cost should be similar with VMC. In our experiments with FermiNet, DiffVMC is slower than VMC because we need to back-propagate through the gradient network. However, this should be mitigated with more adapted network architectures. In general, the major computation bottleneck of QMC is evaluating $\mathrm{tr}(\nabla_{\boldsymbol{x}}\boldsymbol{s}(\boldsymbol{x}))$ which cannot be paralleled efficiently. This is a common problem for all QMC methods. However, by noticing the connection between kinetic energy and score matching, we might be able to accelerate this computation via more efficient score matching techniques (Song & Ermon, 2019; Song et al., 2020; Pang et al., 2020).

## B  PROOF FOR EQUIVARIANCE

We show it for two 1-d particles:

$$\partial_1 \log |\psi(x_1, x_2)| = \lim_{\Delta x \to 0} \frac{\log |\psi(x_1 + \Delta x, x_2)| - \log |\psi(x_1, x_2)|}{\Delta x} \tag{23}$$

$$= \lim_{\Delta x \to 0} \frac{\log |\psi(x_2, x_1 + \Delta x)| - \log |\psi(x_2, x_1)|}{\Delta x} = \partial_2 \log |\psi(x_2, x_1)|. \tag{24}$$

## C  DERIVATION OF THE APPROXIMATED DETAILED BALANCING

The original detailed balancing in Langevin Monte Carlo is

$$P_{\mathrm{acc}}(\boldsymbol{x}'|\boldsymbol{x}) = \frac{\exp\left(-\frac{1}{2\alpha}\|\boldsymbol{x} - \boldsymbol{x}' - \boldsymbol{s}_{\boldsymbol{\theta}}(\boldsymbol{x}')\alpha\|^2\right)\psi_{\boldsymbol{\theta}}(\boldsymbol{x}')^2}{\exp\left(-\frac{1}{2\alpha}\|\boldsymbol{x}' - \boldsymbol{x} - \boldsymbol{s}_{\boldsymbol{\theta}}(\boldsymbol{x})\alpha\|^2\right)\psi_{\boldsymbol{\theta}}(\boldsymbol{x})^2} \tag{25}$$

In our score-based framework, we generally do not have access to wavefunction value. So our goal here is to get rid of the explicit dependencies on $\psi_{\boldsymbol{\theta}}(\boldsymbol{x}')$ and $\psi_{\boldsymbol{\theta}}(\boldsymbol{x})$. We can do so by assuming the score is constant between $\boldsymbol{x}$ and $\boldsymbol{x}'$. We first transform into the log domain:

$$\log P_{\mathrm{acc}} = \underbrace{-\frac{1}{2\alpha}(\|\boldsymbol{x} - \boldsymbol{x}' - \boldsymbol{s}_{\boldsymbol{\theta}}(\boldsymbol{x}')\alpha\|^2 - \|\boldsymbol{x}' - \boldsymbol{x} - \boldsymbol{s}_{\boldsymbol{\theta}}(\boldsymbol{x})\alpha\|^2)}_{A} + \underbrace{2(\log|\psi_{\boldsymbol{\theta}}(\boldsymbol{x}')| - \log|\psi_{\boldsymbol{\theta}}(\boldsymbol{x})|)}_{B} \tag{26}$$

where we use $A$ and $B$ to denote the two components and $\log P_{\mathrm{acc}} = A + B$.

We first simplify $A$. By expanding the norms as $\|\boldsymbol{x} - \boldsymbol{x}' - \boldsymbol{s}_{\boldsymbol{\theta}}(\boldsymbol{x}')\alpha\|^2 = \|\boldsymbol{x} - \boldsymbol{x}'\|^2 - 2\langle\boldsymbol{x} - \boldsymbol{x}', \boldsymbol{s}_{\boldsymbol{\theta}}(\boldsymbol{x}')\rangle\alpha + \|\boldsymbol{s}_{\boldsymbol{\theta}}(\boldsymbol{x}')\|^2\alpha^2$ and $\|\boldsymbol{x}' - \boldsymbol{x} - \boldsymbol{s}_{\boldsymbol{\theta}}(\boldsymbol{x})\alpha\|^2 = \|\boldsymbol{x}' - \boldsymbol{x}\|^2 - 2\langle\boldsymbol{x}' - \boldsymbol{x}, \boldsymbol{s}_{\boldsymbol{\theta}}(\boldsymbol{x})\rangle\alpha + \|\boldsymbol{s}_{\boldsymbol{\theta}}(\boldsymbol{x})\|^2\alpha^2$, we obtain:

$$A = \langle \boldsymbol{x} - \boldsymbol{x}', \boldsymbol{s_\theta}(\boldsymbol{x}') + \boldsymbol{s_\theta}(\boldsymbol{x}) \rangle + \frac{\alpha}{2}(\|\boldsymbol{s_\theta}(\boldsymbol{x})\|^2 - \|\boldsymbol{s_\theta}(\boldsymbol{x}')\|^2) \tag{27}$$

To simplify B, we use the approximation:

$$\log|\psi_{\boldsymbol{\theta}}(\boldsymbol{x}')| - \log|\psi_{\boldsymbol{\theta}}(\boldsymbol{x})| \approx \left\langle \frac{1}{2}\left(\nabla_{\boldsymbol{x}}\log|\psi_{\boldsymbol{\theta}}(\boldsymbol{x}')| + \nabla_{\boldsymbol{x}}\log|\psi_{\boldsymbol{\theta}}(\boldsymbol{x})|\right), \boldsymbol{x}' - \boldsymbol{x} \right\rangle \tag{28}$$

$$= \frac{1}{2}\left\langle \boldsymbol{s_\theta}(\boldsymbol{x}') + \boldsymbol{s_\theta}(\boldsymbol{x}), \boldsymbol{x}' - \boldsymbol{x} \right\rangle \tag{29}$$

Such approximation is commonly employed in finite difference methods. The approximation should be valid since $\boldsymbol{x}$ and $\boldsymbol{x}'$ are close in space. With this approximation, we have:

$$B = 2(\log|\psi_{\boldsymbol{\theta}}(\boldsymbol{x}')| - \log|\psi_{\boldsymbol{\theta}}(\boldsymbol{x})|) \approx -\langle \boldsymbol{x} - \boldsymbol{x}', \boldsymbol{s_\theta}(\boldsymbol{x}') + \boldsymbol{s_\theta}(\boldsymbol{x}) \rangle \tag{30}$$

Finally,

$$\log P_{\text{acc}} = A + B \approx \frac{\alpha}{2}(\|\boldsymbol{s_\theta}(\boldsymbol{x})\|^2 - \|\boldsymbol{s_\theta}(\boldsymbol{x}')\|^2) \tag{31}$$

## D  COMPUTATIONAL SETTINGS

We implement DiffVMC in Pytorch (Paszke et al., 2017) for experiments with Bosons and in Jax (Bradbury et al., 2018) for experiments with Fermions. All gradient computation is done with automatic differentiation from the respective libraries. For experiments with FermiNet, we adapt based on the official implementation of FermiNet (James S. Spencer & Contributors, 2020).

| | | |
|---|---|---|
| Network | Hidden dim | 32 |
| | # MLP layers | 3 |
| Optimization | Optimizer | Adam |
| | Batch size | 256 |
| | Learning rate | 5e-4 |
| | Optimization steps | 2000 |
| | Clip local energy | $5 \times \text{std}$ |
| | WSM - $\beta$ | 1 |
| Langevin dynamics | Langevin dynamics steps | 20 |
| | Metroplis Hasting rejection | Approximate (Equation 9) |
| | Step size $\alpha$ | 0.01 |
| | Score norm clip | 20 |

Table 2: Hyperparameters for Bosons

| | | |
|---|---|---|
| FermiNet | # determinants | 16 (for atoms) or 1 (for QHO) |
| | Single layer hidden dim | 256 |
| | Double layer hidden dim | 32 |
| | # embedding layers | 4 |
| Side network (FFN) | Single layer hidden dim | 64 |
| | Double layer hidden dim | 8 |
| | # embedding layers | 4 |
| Optimization | Pertraining steps | 500 (for B, C) or 1000 (for N, O) |
| | Batch size | 256 (for B, C) or 512 (for N, O) |
| | Optimizer | Kfac |
| | Kfac - damping | 0.001 |
| | Kfac - norm constraint | 0.001 |
| | Initial learning rate | 5e-2 |
| | Learning rate decay | $\text{lr}_{\text{init}} \times (1 + t/10000)^{-1}$ |
| | Clip local energy | $5 \times \text{std}$ |
| | WSM - $\beta$ | 1 or 2 |
| Langevin dynamics | Langevin dynamics steps | 20 |
| | Metroplis Hasting rejection | Approximate (Equation 9) |
| | Step size $\alpha$ | 0.001 or 0.01 |
| | Optimization steps | 200,000 |
| | Score norm clip | 20 |

Table 3: Hyperparameters for Fermions

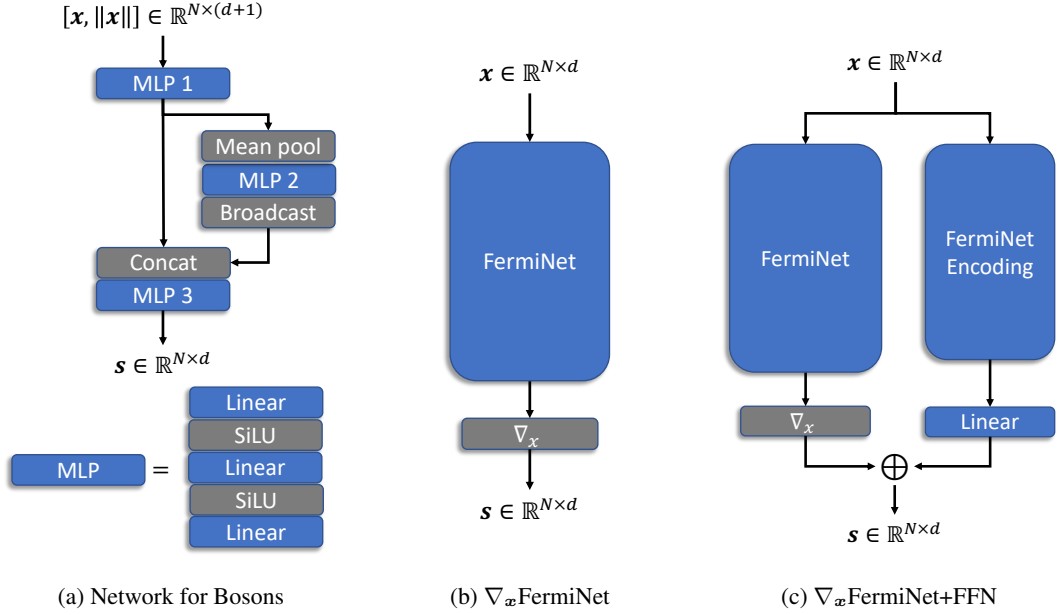

(a) Network for Bosons     (b) $\nabla_x$FermiNet     (c) $\nabla_x$FermiNet+FFN

Figure 3: Network architectures. For simplicity, $-x$ is omitted in $\nabla_x$FermiNet+FFN.

