# OpenReview forum: "A Score-Based Model for Learning Neural Wavefunctions"
_ICLR.cc/2023/Conference — Submitted to ICLR 2023_

### Official Review · Reviewer_afAv · 2022-10-23

**Confidence:** 3
**Correctness:** 4
**Technical Novelty And Significance:** 2
**Empirical Novelty And Significance:** 3
**Recommendation:** 8

**Clarity, Quality, Novelty And Reproducibility:**

**Clarity:**
as a non-expert, I have been able to easily understand the text. The text is clear.


**Quality:** I am not finding the text extremely convincing when it comes to the main claim that modelling the score is ``better" than modelling the wave-function itself. This may be true, in some settings, but I have not finding the text, as well as the numerical simulations, extremely convincing. Again, I may have mis-understood some of the aspects of the proposed methodology. I am looking forward to the authors' response.


**Novelty:** I do not know the modern literature on this problem well-enough to comment.


**Reproducibility:** there are enough details provided in the main text to reproduce the experiments.



**Details Of Ethics Concerns:**

No concern.

**Strength And Weaknesses:**

**Disclaimer:** I am not an expert in ML methods for computing the ground state of quantum systems. I am **not** on top of the current literature.

**Strength:** the paper is accessible and reads reasonably well. I believe that the fundamental motivation for the proposed methodology is the claim that modelling the score of the ground state distribution is somehow easier than modelling the distribution itself (or equivalently, parametrising the score is easier that parametrising the wave function with a neural net). I am finding it interesting that the authors are trying to study this claim. I do not know the literature well enough to comment on the novelty of the approach.

**Weakness:**
1. parametrising the score (instead of the wave function) means that it is not possible to use an exact Metropolis-Hastings (MH) accept/reject. The authors choose to completely ignore the problematic quantities within the MH and I am finding this step a bit hard to swallow. More comments and simulations would be welcome.
2. All the quantities in Equation (4) can be expressed in term of the score: the function $E$ is a function of the score, and the gradient term is exactly the score. Why not discretize this quantity to get an approximation of $\nabla \mathcal{L}(\theta)$ and update the score? Doing this, we do not even need to compute the DMC weights, no?
3. In order to substantiate the claim that modelling the score is better than directly modelling the wave function, the authors should run the method that consists in evolving a population of $N$ particles with the Langevin-dynamics with **exact** accept-reject, with the same computation of the DMC weights, and update of wave function through Equation (4).
4. It is a bit disappointing that the authors are still relying (as they readily admit it) on the parametrisation of the wave-function (i.e. FermiNet parametrisation), which seems to defeat the whole purpose of the whole study.

Again, I would like to re-iterate that I am not an expert in this domain (and may have mis-understood some aspects of the proposed method) and I am looking forward to hearing the authors comments.

**Summary Of The Paper:**

Consider the problem of finding the ground-state of the Hamiltonian $H = -\frac{1}{2} \nabla^2 + V$, i.e. its lowest eigenfunction $\psi$. This corresponds to minimising the loss function $\mathcal{L}(\theta) = \mathbb{E}_{|\psi|^2}\left[ E(x,\psi) \right]$ where $E(x, \psi) = \frac{H \psi(x)}{\psi(x)}$ and the wave-function $\psi$ is parametrised with a neural-network with weight $\theta \in \Theta$. We have that

$$ \nabla \mathcal{L}(\theta) = \mathbb{E}_{|\psi|^2}\left[  \left(E(x,\psi)  -  \mathcal{L}(\theta) \right)  \nabla \log |\psi(x)|  \right].$$

Instead of parametrising the wave function, the articles proposes to parametrize (half-of) the score $s(x) = \nabla \log |\psi(x)|$ of the probability density proportional to $|\psi^2|$. In order to learn the score of the ground-state, the proposed approach maintains at iteration $t \geq 0$ an approximation of the score function $s^{(t)} = \nabla \log |\psi^{(t)}|$ as well as a population of $N$ particles $[x^t_i]_{i=1}^N$ approximately distributed according to $|\psi|^2$.

1. at iteration $t \geq 0$, for a current approximation of the score $s^{(t)}$ and population $[x^t_i]_{i=1}^N$, use (approximate) Langevin dynamics with $s^{(t)}$ and finally compute the usual Diffusion-Monte-Carlo (DMC) weights. This procedure gives a slightly better approximation of the ground state expressed as a weighted population of particles (i.e. it is the working principle of DMC)
2. Instead of resampling with these weights as a standard DMC method would do, use (weighted) score matching to get an approximation go this weighted population of particles. Since the weighted population of particles was a better approximation of the ground state, this results in a better estimate of the score of the ground state.
3. iterate



**Summary Of The Review:**

In many settings, modelling the score of a distribution has advantages when compared to modelling the distribution itself (eg. generative modelling). The text proposes to investigate this claim in the context of finding the ground state of the Schrodinger equation.

After reading the text, it is not entirely clear to me whether modelling the score has advantages over modelling the wave-function itself. Still,  the text seems to demonstrate that this approach is viable and deserves more numerical investigations.

---

> ### Author Response · Authors · 2022-11-18
> **Response to Reviewer afAv (Part 2)**
>
> **Quality.**
> I am not finding the text extremely convincing when it comes to the main claim that modeling the score is "better" than modeling the wave-function itself. This may be true, in some settings, but I have not found the text, as well as the numerical simulations, extremely convincing. Again, I may have mis-understood some of the aspects of the proposed methodology. I am looking forward to the authors' response.
>
> **A.** As our motivation, using score is more fundamental to energy minimization (as in local energy formula). But currently we cannot claim that it is better. However, our method has the unique ability to optimize parameterized score toward ground states.
>
> **Summary.**
> After reading the text, it is not entirely clear to me whether modeling the score has advantages over modeling the wave-function itself. Still, the text seems to demonstrate that this approach is viable and deserves more numerical investigations.
>
> **A.** This is correct. We aim to design a score-based model in the style of VMC. This could potentially lead to more expressive Ansatz. The current focus of deep-learning-based QMC is mostly on fermion systems. However, due to the difficulty of fermion sign problem, designing effective networks to solve the sign problem is not straightfoward. So we would like to first validate the optimization method with feasible solutions. We hope more adapted networks and optimization methods could be developped in future research.
>
> [1] Wilson, Max, et al. "Simulations of state-of-the-art fermionic neural network wave functions with diffusion Monte Carlo." arXiv preprint arXiv:2103.12570 (2021).
>
> [2] Hermann, Jan, Zeno Schätzle, and Frank Noé. "Deep-neural-network solution of the electronic Schrödinger equation." Nature Chemistry 12.10 (2020): 891-897.

---

> > ### Comment · Reviewer_afAv · 2022-11-21
> > **clarifications**
> >
> > Thank you very much for your detailed answer.
> >
> > > More generally, we cannot directly use the sample average of $\mathcal{L}(\theta)$ as loss function. This is because  $\mathcal{L}(\theta)$ depends on the distribution of samples (wavefunction). When we minimize  $\mathcal{L}(\theta)$ for the current batch of samples, the energy for the current samples is decreased, but the underlying distribution will also be changed. Since the kinetic part of the local energy depends explicitly on (score of) distribution, the energy expectation may not decrease. Hence, when computing the gradient, we must consider the distribution shift. This is why we need to use the energy gradient in VMC.
> >
> > I am not sure I understand very well. So suppose you have a population of particles $x_i$ approximating the current $\psi_\theta^2$, and a current approximation $s_\theta$ of the score. One can approximate $\mathcal{L}(\theta) \approx \widehat{\mathcal{L}}(\theta)  = (1/N) \sum_{i=1}^N E_i(x_i, \theta)$ since $E_i(x_i, \theta)$ can be computed from the score (and its derivative) only. Then, can't we take a gradient step (i.e. $\nabla_{\theta} \widehat{\mathcal{L}}(\theta)$) to slightly update the approximation of the score, and then update the particles with Langevin dynamics (as done in the article) with this updated score to make sure that the particles are approximately distributed according to the updated distribution.

---

> > > ### Author Response · Authors · 2022-11-21
> > > **Response**
> > >
> > > Thank you for your reply. In the following we show why using the sample mean of local energies as loss function will not work.
> > >
> > > We can write the local energy at $x_i$ as $E_L(x_i, \theta)=E_k(x_i, \theta)+V(x_i)$, where $E_k$ is the kinetic energy and $V$ is the potential energy. Note that only $E_k(x_i, \theta)$ depends on parameters $\theta$ whereas $V(x_i)$ only depends on the positions $x_i$ and is independent of the wavefunction or $\theta$. Therefore, if we use the sample mean of local energies as loss function, then the parameter updates will be independent of potential $V$. Moreover, since Langevin dynamic (or MCMC) does not involve $V$ neither, the entire optimization process will be independent of $V$, which will make optimization fail.
> > >
> > > Also, having noticed the connection between kinetic energy and score matching objective, we can have another interpretation from score matching. As discussed above, since $V(x_i)$ does depends on $\theta$, optimizing $\hat{\mathcal{L}}(\theta)$ is equivalent to optimizing $\frac{1}{N}\sum_i E_k(x_i, \theta)$, which according to our observation, is equal to $-\frac{1}{4}\frac{1}{N}\sum_i ISM(x_i, \theta)$, where $ISM$ is the implicit score matching objective. So intuitively, optimizing $\hat{\mathcal{L}}(\theta)$ is equivalent to optimizing the score matching objective in the opposite direction, which will make the learned score (or wavefunction) away from sample distribution. On the other hand, Langevin dynamics (or MCMC) will make sample distribution close to the learned score (or wavefunction). As a result, the optimization process will never attain equilibrium.
> > >
> > > We hope these clarify the motivation for a more complicated loss. We will include these discussions in the text.

---

> > > > ### Comment · Reviewer_afAv · 2022-11-22
> > > > **Thanks**
> > > >
> > > > Indeed, thanks, this clarifies everything.

---

> > > > > ### Author Response · Authors · 2022-11-22
> > > > > **Thank you**
> > > > >
> > > > > Thank you again for your review. We are grateful for your valuable feedback to improve the paper!

---

> ### Author Response · Authors · 2022-11-18
> **Response to Reviewer afAv (Part 1)**
>
> Thanks a lot for your review. We reply to you comments below:
>
> **W1.**
> Parameterizing the score (instead of the wave function) means that it is not possible to use an exact Metropolis-Hastings (MH) accept/reject. The authors choose to completely ignore the problematic quantities within the MH and I am finding this step a bit hard to swallow. More comments and simulations would be welcome.
>
> **A.** It is true that we cannot compute the exact MH ratio without the access to the wavefunction. We are aware of this quantity and we proposed a way to approximate this term in eq.10, section 3.2. In the revised text we have added detailed derivation of this approximation in section 3.2 and Appendix C. We essentially use the assumption that the average gradient between $\log|\psi(x)|$ and $\log|\psi(x^\prime)|$ can be approximated by $\frac{1}{2}(\nabla_x\log|\psi(x)|+\nabla_x\log|\psi(x^\prime)|)$. This assumption should be valid since $x$ and $x^\prime$ are close. The approximated MH rejection is used in all experiments. When $\alpha=1e-3$, the acceptance ratio is about 0.996 to 0.999, which is consistent with DMC [1]. To our best knowledge, this approximated MH step is new and we agree that it deserves more numerical study. However, to better examine its effect, we may need to first achieve higher accuracies with score-based models.
>
>
> **W2.**
> All the quantities in Equation (4) can be expressed in term of the score: the function $E$ is a function of the score, and the gradient term is exactly the score. Why not discretize this quantity to get an approximation of $\nabla \mathcal{L}(\theta)$ and update the score? Doing this, we do not even need to compute the DMC weights, no?
>
> **A.** The gradient term in the energy loss $\nabla_\theta\log|\psi_\theta(x)|$ is w.r.t. the parameters. However, the score defined in our method is $\nabla_x\log|\psi_\theta(x)|$ is w.r.t. the input. So the gradient term is not the score.
>
> More generally, we cannot directly use the sample average of $\mathcal{L(\theta)}$ as loss function. This is because $\mathcal{L(\theta)}$ depends on the distribution of samples (wavefunction). When we minimize $\mathcal{L(\theta)}$ for the current batch of samples, the energy for the current samples is decreased, but the underlying distribution will also be changed. Since the kinetic part of the local energy depends explicitly on (score of) distribution, the energy expectation may not decrease. Hence, when computing the gradient, we must consider the distribution shift. This is why we need to use the energy gradient in VMC.
>
>
> **W3.**
> In order to substantiate the claim that modeling the score is better than directly modeling the wave function, the authors should run the method that consists in evolving a population of N particles with the Langevin-dynamics with exact accept-reject, with the same computation of the DMC weights, and update of wave function through Equation (4).
>
> **A.** If we understand correctly, you describe the experiment where we train Ferminet with energy gradient while using Langevin Monte Carlo as sampling method. To our understanding, this a standard way to replace MCMC sampling and is used in PauliNet[2]. MCMC should be good enough for this task. In FermiNet, only the result for MCMC is reported. We think the comparison between MCMC and Langevin Monte Carlo is not the main problem in this study. Instead we aim to design a framework to enable the possibility to directly optimize the parameterized score. Please refer to section 5 in the revised text.
>
> **W4.**
> It is a bit disappointing that the authors are still relying (as they readily admit it) on the parameterization of the wave-function (i.e. FermiNet parameterization), which seems to defeat the whole purpose of the whole study.
>
> **A.** We agree that still relying on FermiNet appears to be a limitation. We have made attempts to design purely score-based Ansatz. But so far they are unsuccessful due the diverging behavior at nodal surface. In fact, the fermion sign problem is a long standing challenge. Currently we are not sure whether a purely score-based model is feasible for fermions.
> In the revised paper, we also explore the possibility to augment the Ferminet gradient with a feed-forward network (FFN). In this way, the fermion constraint is satisfied with FermiNet gradient and the FFN could potentially refine the predicted score. We show that this new architecture works for fermions in QHO. Unfortunately using FFN does not bring much improvement to atomic systems. However, such optimization cannot be achieved without our score-based method. We hope the capacity of our method on fermion systems can be further explored in future research.

---

### Official Review · Reviewer_Lxrt · 2022-10-24

**Confidence:** 3
**Correctness:** 3
**Technical Novelty And Significance:** 3
**Empirical Novelty And Significance:** 2
**Recommendation:** 6

**Clarity, Quality, Novelty And Reproducibility:**

Clarity:

The paper is generally easy-to-follow, but has some mathematical typos as I mentioned in the above section. The main topic of this paper, i.e., the connection between the local energy and ISM, is not entirely clear for its current form.

Quality:

The motivation of this paper is clear. However, experiments do not support the motivation (by-passing the wavefunction evaluation and enhancing the computational efficiency)

Novelty:

An introduction of the score matching for QMC is novel for me. However, my evaluation might be not exhaustive because I am not an expert on QMC field.

Reproducibility:

The paper contains an incomplete set of used model architectures and hyper-parameters. Code is not made publicly available.

**Strength And Weaknesses:**

Strength:

1. The motivation of the propsed framework, i.e., learning and sampling ground states by using the score function only, is clear. First, MCMC step of VMC learning can be removed. It takes a similar advantage as by-passing expensive MCMC likelihood estimation in the score matching method of EBMs. Second, as the authors mention, one can omit the necessity of computing the score function from the wavefunction.

2. The authors introduce a permutation-equivariant architecture [1] to guarantee the indistinguishability of quantum mechanical particles. To me, it is a novel application of [1] (although not sure because I am not an expert of machine-learning-based quantum simulations).

Weakness:
1. The authors derive the local energy of QMC, and conclude its kinetic term is equal to the implicit score matching (ISM) [2] (up to a constant). However, it is not true considering a coefficient of the norm of the score function $||s_{\theta}(\mathbf{x})||^2$. Note that the kinetic term of the local energy of QMC is given by $\text{tr}(\nabla_{\mathbf{x}}s_{\theta}(\mathbf{x})) + ||s_{\theta}(\mathbf{x})||^2$ while the original ISM is $\text{tr}(\nabla_{\mathbf{x}}s_{\theta}(\mathbf{x})) + 0.5||s_{\theta}(\mathbf{x})||^2$. Thus, it is not entirely clear the connection between the local energy and ISM.

2. It seems that the major advantage of the proposed DVMC is its learning and sampling efficiency. However, there is no benchmark on the computational efficiency of DVMC.

3. The authors introduce four assumptions (in page 5) to eliminating the wavefunction-dependency and distributional shift of the porposed framework. However, they do not dicuss whether the introduced assumptions are reasonable (acceptable) or not.

4. For fermion systems, the proposed method requires computing the wavefunction. Therefore, the main claim of this paper is partially applicable (only for boson systems).

5. Experimental results are weak. The harmonic trap example is very simple and can be solved analytically. Atomic systems (B/C/N/O) are more intersting, but they are fermion systems, which doe not support the main claim of this paper as I mentioned in 5.

6. The authors calim "numerous examples in machine learning have shown that implicitly modeling the data distribution provides better representation" and "properly chosen latent embedding can better describe the struvture of the data space" with refering VAE and GAN literatures in page 1 of the paper. Because the used algorithm (score matching) is not based on the latent representation and the used data (quantum state) seems to be not an embedded manifold, such claims are not entirely clear to me.

Some minor issues:
1. In (3), it seems that $\mathbf{x} \sim \psi_{\theta}^{2}(\mathbf{x})$ should be $\mathbf{x} \sim \psi_{\theta}^{2}(\mathbf{x}) / \int \psi_{\theta}^{2}(\mathbf{x}) d\mathbf{x}$.
2. In (7), the ISM should be $\mathbb{E} \[ \text{tr}(\nabla_{\mathbf{x}}s_{\theta}(\mathbf{x})) + 0.5||s_{\theta}(\mathbf{x})||^2 \]$.
3. In page 6, the authors mention "The equivariance can be easily achieved by considering the input coordinates as a set". Please elarborate it more detaily or add a reference (might be [1]?)
4. The potential $V(\mathbf{x})$ is not included in (18 - 19).

** *
[1] Zaheer, M., Kottur, S., Ravanbakhsh, S., Poczos, B., Salakhutdinov, R. R., & Smola, A. J. (2017). Deep sets. Advances in neural information processing systems, 30.

[2] Hyvärinen, A., & Dayan, P. (2005). Estimation of non-normalized statistical models by score matching. Journal of Machine Learning Research, 6(4).


**Summary Of The Paper:**

The authors propose Diffusion Variational Monte Carlo (DVMC) for simulating many-body quantum systems based on the neural score function. Contrast to a conventional Variational Monte Carlo (VMC), the proposed method requires estimating the score function only, which is a gradient of the logarithm of the wavefunction, rather than evaluating the wavefunction itself. To this end, the authors formulate a new loss function, named weight score matching (WSM, which is analogous to the score matching used for energy-based models (EBMs)), that can learn a ground state of the target wavefunction with the score function only. By combinding such a loss function with the Lagevin dynamics, the proposed DVMC framework samples and learns the targeted ground state simultaneuosly. The authors validate their approach for two quantum systems.

**Summary Of The Review:**

Although the paper is interesting, I think there is still much room for improvement. I think the paper is not ready for publication in its current form.

---

> ### Author Response · Authors · 2022-11-18
> **Response to Reviewer Lxrt (Part 3)**
>
> **Minors.**
>
> 1. In (3), it seems that x2(x) should be $x\sim \psi_\theta^2(x)$ should be $x\sim      \psi_\theta^2(x)/\int\psi_\theta^2(x)$.
>
>     **A.** You are right. We have corrected in the revised text. Thanks for catching it.
>
> 2. In (7), the ISM should be $\mathbb{E}[tr(\nabla_x s(x))+0.5\|s(x)\|^2]$.
>
>     **A.** Please refer to our response to W1.
>
> 3. In page 6, the authors mention "The equivariance can be easily achieved by considering the input coordinates as a set". Please elaborate it in more details or add a reference (might be [1]?)
>
>     **A.** We have included the related references in the revised version. We also include figures for architectures in Figure 3 in Appendix.
>
> 4. The potential V(x) is not included in (18 - 19).
>
>     **A.** $V(x)$ is in the local energy $E_L(x)$. We have included it explicitly in the revised version (eq.15).
>
>
> **Clarity.**
> The paper is generally easy-to-follow, but has some mathematical typos as I mentioned in the above section. The main topic of this paper, i.e., the connection between the local energy and ISM, is not entirely clear for its current form.
>
> **A.** We have fixed the typos in the WSM formula that might cause confusion. Please refer to W1 for detailed explanation.
>
> As mentioned in W1, the connection between local energy and ISM currently only provides computational convenience. However, we believe that there should be a deeper connection between these two. For example, in DMC, the short-time approximation is directly related to the exact form of the kinetic energy. In the revised text, we have improved the derivation of WSM in section 3.3 but currently our connection to DMC is not very rigorous so we cannot claim the connection between local energy and ISM is fundamental to our method. Nonetheless, the connection indeed exists and could be interesting for future research.
>
>
> **Quality.**
> The motivation of this paper is clear. However, experiments do not support the motivation (by-passing the wavefunction evaluation and enhancing the computational efficiency)
>
> **A.** Please refer to section 5 and Appendix A in the revised text.
>
>
> **Reproducibility.** The paper contains an incomplete set of used model architectures and hyper-parameters. Code is not made publicly available.
>
> **A.** We have included the implementation details in Appendix D. We will release our code after the review process.

---

> > ### Comment · Reviewer_Lxrt · 2022-11-30
> > **Reply to the authors' response**
> >
> > I appreciate the authors’ thorough response to my questions.
> >
> > Especially, I am happy to know that the revised version of the paper is cosiderably improved regarding the motivation and clarity.
> >
> > After reading the authors' response, now I think the paper’s contribution outweighs my initial concerns.
> >
> > I increased my review score accordingly.

---

> > > ### Author Response · Authors · 2022-11-30
> > > **Thank you**
> > >
> > > Thank you again for your review! We greatly appreciate your detailed feedback and openness to adjusting scores!

---

> ### Author Response · Authors · 2022-11-18
> **Response to Reviewer Lxrt (Part 2)**
>
>
>
> **W4.**
> For fermion systems, the proposed method requires computing the wavefunction. Therefore, the main claim of this paper is partially applicable (only for boson systems).
>
> **A.** Compared to VMC and DMC, the major difference of the proposed method is the ability to optimize parameterized score. We add experiments with a new architecture where the score is modeled as FermiNet gradient + a feed-forward network (FFN). Under this setting, the fermion constraint is handled by FermiNet gradient and the FFN could further refine the score prediction. We show that this architecture works correctly for fermions in QHO and for atoms. Although the accuracy on atoms are not improved by FFN, such optimization cannot be achieved without the proposed score-based optimization.
>
> Antisymmetry for fermions has been studied for long time. For example, FermiNet is based on the Slater determinant. We are not sure whether a purely score-based method for fermions is feasible or not. So currently fermions may not be the best choice to showcase the strength of our method. However, we think it is sufficiently complex to show that our score-based optimization can correctly find ground states. Recent advancements in deep QMC also make it a good test bed. Nevertheless, we agree that partially requiring an energy-based model appears to be a limitation. One justification could be that, as the motivation of our additional experiments, combining energy-based model and score-based model could potentially improve the expressiveness of Ansatz. We are not able to claim this by now and we hope our work can bring insights in future research.
>
>
> **W5.**
> Experimental results are weak. The harmonic trap example is very simple and can be solved analytically. Atomic systems (B/C/N/O) are more interesting, but they are fermion systems, which does not support the main claim of this paper.
>
> **A.** Since the proposed framework is new, we would like to first validate it on systems where the ground state is exactly known. This is the motivation of validating on harmonic trap. In the revised version, we also include fermions in QHO, which is more challenging than bosons.
>
> For atoms, please refer to our response to W4.
>
>
> **W6**
> The authors claim "numerous examples in machine learning have shown that implicitly modeling the data distribution provides better representation" and "properly chosen latent embedding can better describe the structure of the data space" with referring VAE and GAN literatures in page 1 of the paper. Because the used algorithm (score matching) is not based on the latent representation and the used data (quantum state) seems to be not an embedded manifold, such claims are not entirely clear to me.
>
> **A.** We agree that our method is not related to latent representation. We use examples of VAE and GAN only to show that implicit modeling is more promising than explicit modeling. We have removed the second cited sentence in the revised text.

---

> ### Author Response · Authors · 2022-11-18
> **Response to Reviewer Lxrt (Part 1)**
>
> Thanks a lot for your review. We reply to you comments below:
>
> **W1.**
> The authors derive the local energy of QMC, and conclude its kinetic term is equal to the implicit score matching (ISM) [2] (up to a constant). However, it is not true considering a coefficient of the norm of the score function $\|s(x)\|^2$. Note that the kinetic term of the local energy of QMC is given by $\text{tr}(\nabla_x s(x))+\|s(x)\|^2$ while the original ISM is $\text{tr}(\nabla_x s(x))+0.5\|s(x)\|^2$. Thus, it is not entirely clear the connection between the local energy and ISM.
>
> **A.** Because $p(x)=\psi(x)^2$, the score defined in our paper is half of the score as defined in the original score matching. If we instead let $s(x)=\nabla_x\log p(x) = 2\nabla_x\log|\psi(x)|$ as in the original score matching, we will have $\nabla_x\log|\psi(x)| = \frac{1}{2}s(x)$. Based on Equation (4) in the revised text, the kinetic term would be $-\frac{1}{2}\sum_i \left( \frac{\partial^2 \log |\psi(x)|}{\partial x_i^2}+\left(\frac{\partial \log|\psi(x)|}{\partial x_i}\right)^2 \right) =  -\frac{1}{2}\left(\text{tr}(\nabla_x \frac{1}{2} s(x)) + \|\frac{1}{2} s(x)\|^2\right)=-\frac{1}{4} \left(\text{tr}(\nabla_x s(x))+ 0.5 \|s(x)\|^2\right)=-\frac{1}{4}ISM$. The $0.5$ before $\|s(x)\|$ appears because of the square of the norm. In our paper, by abuse of notation we define $s(x)\leftarrow \frac{1}{2}s(x)=\nabla_x \log |\psi(x)|$. We do so in order to simplify the description and because the physical meaning of $\nabla_x\log|\psi(x)|$ is the same with the original score. We highlight this point before section 3.1 in the revised text.
>
> There was indeed a typo in the WSM loss where we used the original score definition. We have fixed this typo in the revised text. We apologize if confusion was caused.
>
> Currently this connection between kinetic energy and ISM is only used to avoid repeated computation. However, there could be deeper connections.
>
> **W2.**
> It seems that the major advantage of the proposed DVMC is its learning and sampling efficiency. However, there is no benchmark on the computational efficiency of DVMC.
>
> **A.**
> Compared to VMC and DMC, the major difference of the score-based model is its ability to optimize the parameters in the score function. This is not achievable by VMC or DMC. We agree that the comparison was not clearly presented. In the revised text we discuss it in more details. Please refer to section 5 in the revised text.
>
> We do not claim that our method currently has better computational efficiency than VMC or DMC. We describe the computational efficiency in Appendix A of the revised text.
>
>
> **W3.**
> The authors introduce four assumptions (in page 5) to eliminate the wavefunction-dependency and distributional shift of the proposed framework. However, they do not discuss whether the introduced assumptions are reasonable (acceptable) or not.
>
> **A.** We add detailed derivation for the approximated MH rejection in section 3.2 and appendix C of the revised text. We essentially use the assumption that the average gradient between $\log|\psi(x)|$ and $\log|\psi(x^\prime)|$ can be approximated by $\frac{1}{2}(\nabla_x\log|\psi(x)|+\nabla_x\log|\psi(x^\prime)|)$. This assumption should be valid since $x$ and $x^\prime$ are close. However, it is true that more experiments under different settings are needed to fully quantify the approximation.
>
> For the WSM objective. We agree that the original derivation was not well justified. We tried to derive a rigorous proof to it but was not successful. Nevertheless, we are able to make a better connection from imaginary time evolution. We also prove that the loss is unbiased. Please refer to section 3.3 in the revised text.

---

### Official Review · Reviewer_FLe5 · 2022-10-24

**Confidence:** 4
**Correctness:** 4
**Technical Novelty And Significance:** 3
**Empirical Novelty And Significance:** 2
**Recommendation:** 5

**Clarity, Quality, Novelty And Reproducibility:**

The paper is clearly written and explains relevant concepts well. The article includes relevant references and citations. I was not able to locate all of the details to assess how reproducible the results are (presumably in the appendix which was not included with the main pdf?). Code availability is not stated.

**Strength And Weaknesses:**

Strengths:
* The paper studies an underrepresented approach to modeling quantum states based on gradients of the logarithm of the wave function amplitude.
* Paper explains methodology clearly and accurately

Weaknesses:
* The proposed method seems to be most suited for problems without the “sign problem”. In this setting diffusion monte carlo and similar methods already provide very good results and then at the very least should be considered as a baseline.
* Some approximations made to set up the optimization loop are not fully justified (or I might have missed the argument, e.g. eqs. 14->15;)


**Summary Of The Paper:**

The paper proposes a method for solving the quantum many body problem using neural networks as variational ansatz. In contrast to more standard approaches which model the wave function directly, authors use a score-based model (hence modeling the gradient of the underlying probability distribution associated with the wavefunction). Authors draw on techniques from score based modeling and diffusion Monte Carlo to formulate a schema for training such models. They demonstrate that the proposed approach works well for the bosonic quantum harmonic trap. Authors also achieve good results for fermionic atomic systems by using gradients of a FermiNet as a model for the score function.

**Summary Of The Review:**

I enjoyed reading the paper and I think the direction taken by the authors is both promising and interesting. Results and ideas are presented clearly and concisely. Authors honestly point out challenges associated with nodal structure in fermionic systems.

While I’m very excited about the field and the research direction considered in this paper, I’m a bit concerned that the proposed method addresses problems that are already “solved” by classical techniques such as diffusion Monte Carlo (DMC). In most cases the resort to parametric variational Monte Carlo (VMC) methods is motivated by the sign problem which results in poor estimates in DMC techniques. This is however an exciting direction and would be of great interest to a more narrow audience in computational physics (and a potentially major milestone if such method could be effectively generalized to fermionic systems). Hence I’m leaning towards the score of (5), but could be convinced otherwise.

---

> ### Author Response · Authors · 2022-11-18
> **Response to Reviewer FLe5**
>
>
> Thanks a lot for your review. We reply to you comments below:
>
> **W1.**
> The proposed method seems to be most suited for problems without the “sign problem”. In this setting diffusion monte carlo and similar methods already provide very good results and then at the very least should be considered as a baseline.
>
> **A.**
> The fermion sign problem indeed poses a challenge for the score-based model due to the antisymmetry constraint to the underlying wavefunction. One possibility is to use the gradient of an antisymmetric wavefunction Ansatz, such as the FermiNet in the paper. During rebuttal, we explore the possibility to augment this gradient with output from an additional output from an feedforward network. Although the results are not improved in this experiment, we show that in such a way, our score-based model can go beyond the wavefunction model and could potentially gain in expressiveness.  Another possibility is to get inspiration from DMC and add the antisymmetry directly into score. We found this to be difficult due to the divergence at fermion nodes but we do not rule out this possibility. We think eventually our method could be able to handle the sign problem,
>
> The presented method is suitable for modeling quantum system in continuous space. Currently, deep-learning-based VMC/DMC in continuous space are mostly applied to quantum chemistry. In which case we model the electrons, which are fermions thus give rise the sign problem. There are other applications without the sign problem such as cold atom systems but deep learning settings for such problems are still under explored. Applying our methods to such problems could be challenging. For discrete case, there are deep learning methods applied to condensed matter physics where the systems are modeled as lattices (as mentioned in introduction). Our method is currently not suitable for such problems but could be extended in future research.
>
>
> **W2.**
> Some approximations made to set up the optimization loop are not fully justified (or I might have missed the argument, e.g. eqs. 14 - 15)
>
> **A.** We add more details for the derivation of the approximated rejection in appendix C of the revised paper.
>
>
> **Clarity.**
> I was not able to locate all of the details to assess how reproducible the results are (presumably in the appendix which was not included with the main pdf?). Code availability is not stated.
>
> **A.** We add our computational settings for the experiments in appendix D of the revised paper. We will release our code after the review process.
>
>
> **Summary.**
> I’m a bit concerned that the proposed method addresses problems that are already “solved” by classical techniques such as diffusion Monte Carlo (DMC). In most cases the resort to parametric variational Monte Carlo (VMC) methods is motivated by the sign problem which results in poor estimates in DMC techniques.
>
> **A.**
> The unique value of our approach is the ability to optimized the parameterized score, which appears to be more fundamental to energy minimization. This is not achievable by DMC or VMC. We discuss the relation to DMC and VMC in section 5 of the revised paper.
>
> It is true that the presented approach would be more applicable if there is solution to the fermion sign problem. We explore the simplest solution where antisymmetry is still handled globally. We hope that the proposed score-based framework could bring new insights to deep-learning-based QMC and eventually go beyond VMC/DMC on certain practical problems, whether in terms of expressiveness or efficiency.

---

> > ### Comment · Reviewer_FLe5 · 2022-11-28
> > **Reply**
> >
> > I would like to thank the authors for their reply and improvements to the text.
> >
> > I find the additional comments and paragraphs added to the main text clarifying and helpful. While I'm still very excited about the proposed approach, I'm not convinced that the value of optimizing the parameterized score is well supported. Hence I'm keeping my rating as is, unless convinced by other reviewers.

---

> > > ### Author Response · Authors · 2022-11-30
> > > **[Part 1] Thanks for your reply**
> > >
> > > Thank you for your reply. We sincerely appreciate your effort in evaluating our work. We believe your understanding of the paper is accurate and your feedback is highly valuable.
> > >
> > > We do believe that the proposed approach is of value. The ultimate goal of optimizing parameterized scores could be eliminating the use of Slater determinants or any other globally antisymmetric functions. These functions could be difficult to optimize and their computation could be expensive (e.g. an adapted KFAC optimizer is needed to optimize FermiNet and computing determinant takes $O(N^3)$). One possible way to do this could be using some "cancellation" mechanism, such as in the released node method [1] in DMC. In such methods, we allow samples to have both positive and negative weights. Consequently, positive and negative samples can cancel each other such that the resulting weighted distribution could be equivalent to an antisymmetric distribution. In DMC, since scores cannot be optimized, negative weights have to be kept by walkers during the entire optimization which could potentially complicate the optimization and increase variance. In contrast, our approach enables the optimization of parametrized scores, so we can make the weighted distribution converge to the true antisymmetric distribution.
> > >
> > > As pointed out by reviewer **afAv**, currently our paper mainly shows that the proposed score-based optimization framework is a viable solution to optimize quantum systems toward ground states. Our additional experiment in rebuttal also show that by optimizing scores we can go beyond wavefunction-based Ansatzes. However, the addition of the side network does not improve the results so we agree that it is not very convincing. We might move the additional experiment to the appendix in future versions.
> > >
> > > Despite these possible directions, it could be challenging to implement them and truely demonstrate the value of our approach. During the course of this work, we have made considerable effort to develop purely score-based methods (e.g. using "cancellation"). However, the antisymmetric property could not be fully perserved, even for the simple QHO setting. This is potentially due to the discontinuity issued as mentioned in our paper. On the other hand, using the gradients of globally antisymmetric functions is a feasible solution. According to our experments, the optimization through the gradient network seems to slightly impede the convergence hehavior, so more suitable optimizator could be in need. However, as also pointed out by other reviewers, the advantage of using gradient network is not very clear since we still need to somehow evaluate wavefunctions. In this work, we test the gradient network approach on atomic systems mainly to validate the correctness of our the proposed optimization framework.
> > >
> > > With above considerations, in this paper, we focus on the score-based optimization framework itself and present it as a milestone toward more practical solutions. We think our approach is a principled framework that can optimize effectively toward ground states, although deeper theoretical garantees could be developped. We will continue developping the proposed approach in future works and hopefully the challenging fermion sign problem could be resolved with the above mentioned methods (or a mixture of both).
> > >
> > >
> > > [1] Ceperley, David M., and Berni J. Alder. "Ground state of the electron gas by a stochastic method." Physical review letters 45.7 (1980): 566.

---

> > > ### Author Response · Authors · 2022-12-02
> > > **[Part 2] Additional results and failure cases for Fermions with pure score functions**
> > >
> > > To make it more concrete, we would like to show some additional results with a score network developed in the early stage of this work, where we tried to use pure score functions to learn ground states for fermions in QHO potentials.
> > >
> > > In 1D, fermion wavefunctions become zero exactly when any two particles coincide in space. To handle such antisymmetric constraint, we can let scores diverge at such configurations. To be concrete, the score for the $i$-th particle is defined as $s_\theta(x_i,|x_1,\dots,x_N)=\text{FFN}(x_i|x_1,\dots,x_N)+\sum_{j\neq i}\frac{1}{x_i-x_j}$, where FFN is a feedforward network with the same architecture used for the Bosons. The second term in the right hand side ensures the score to diverge when any two particles are close. The results are shown [in this plot](https://figshare.com/s/babe327fb481fcfad8ca) (please open in a new tab). The single-particle energy levels are $n+\frac{1}{2}, n=0,1,2,\dots$. So by Pauli exclusion, the ground states energies for 2 and more particles are 2, 4.5, 8, 12.5, etc. We can see that this works out correctly for the 1D case. However, this is not surprising because the ground truth scores function for 1D QHO are in the form of $s_{\text{gt}}(x_i|x_1,\dots,x_N)=-x_i+\sum \frac{1}{x_i-x_j}$. So the learning is not difficult as long as the optimization is correct. We show the sample distributions for 2 particles [in this plot](https://figshare.com/s/6a5a6614c9a6980ef41e), the antisymmetric w.r.t. to $x_1=x_2$ is correctly captured.
> > >
> > > This however does not extend to more than 1D. The results for the 2D case with the same network is shown in [this plot](https://figshare.com/s/7b7e7bcd4d110ca79907). The 2D single-particle energy levels are 1, 2, 2, 3, 3, 3, 4, 4, 4, 4, ... (note degenerencies in energy levels). So the ground states energies for 2 and more particles should be 3, 5, 8, 11, 14, etc, as correctly predicted in Figure 2(b) of our paper. However, the prediction with the pure score function gives 4, 9, 16, 25, 36, etc. So this simple network fails to capture the 2-dimensional fermion nodes and constraints are too restrictive. With more sophisticated network designs or training strategies we are able to get correct energies for 2 particles, but still fail for more general cases. In fact, fermion nodal surfaces for 2 or higher dimensions can be arbitrary and can occur without contact between particles. For example, as depicted in [this plot](https://figshare.com/s/b82e6b7c6e7effee4924), two fermions can exchange their positions with a 180-degree rotation around their center. The wavefunction will change sign before and after the rotation so the wavefunction must be zero at least once during the rotation, however, the distance between two particles do not change. With these observations we conclude that designing antisymmetric score functions without using gradients of determinants is highly non-trivial. Deeper insights are needed to learn antisymmetric scores in higher dimensions.
> > >
> > > We will include these discussions in the appendix of future versions. Hope they improve the clarity of motivations and challenges of the proposed framework.

---

### Official Review · Reviewer_PhHJ · 2022-10-25

**Confidence:** 3
**Correctness:** 3
**Technical Novelty And Significance:** 3
**Empirical Novelty And Significance:** 2
**Recommendation:** 6

**Clarity, Quality, Novelty And Reproducibility:**

Suggestions for making the paper more accessible to those without physics expertise:

1. The relation between the WSM objective and V could be explained intuitively to help reader understanding. From what I gather, the score model initially points in many random directions, and the weightings gradually guide the score function to develop "local minima" (for vector fields, this would mean regions where the score function "points to") where the potential V is minimized by causing the model to gradually see many samples of states with lower potential energy. Is it correct to say that the potential V affects learning only through the weightings? It would be helpful to compare the role of V in the proposed work and existing works that have an explicit wavefunction.
2. The exact objective of the method could be stated more clearly. The second-to-last paragraph of page 2 starts with "Our objective is to find the ground state $\psi_0$) but the method appears to learn the gradient of the ground state and samples from the ground state. What is the exact primary goal? I suppose that samples from the ground state wavefunction are the practical objective. If the goal is simply to draw samples from the minimum-energy ground states, why not optimize $V(x)$ directly? These questions are likely naive but could help other non-experts.

**Strength And Weaknesses:**

STRENGTHS:

1. The work is based on an interesting insight into the functional form of the wavefunction that appears in the local energy loss. Replacing the gradient of an explicit wavefunction with a function that directly estimates the score function is a compelling proposal that is aligned with similar developments in generative modeling.
2. The hybridization of DMC and VMC is an interesting solution to the optimization difficulties that arise when replacing the explicit wave function with the score function. DMC provides a way to move towards states with low potential, while VMC provides a more principled probabilistic proposal for DMC based on the score matching objective.
3. The work explores an exciting area at the intersection of deep learning and computational physics.

WEAKNESSES:

3. The Ferminet network used for experiments already provides an estimate for $\psi_0$, so the motivation for score-based modeling is not as clear. What is the motivation or benefit of learning the gradient of a scalar-valued net, instead of learning a network that outputs the estimated gradient directly as typically done in score-based modeling? The fermion restrictions are a difficult challenge with limited options available and the proposed method might be useful for future architectures, but at the present it seems roundabout to use score-based modeling for a scalar valued net.
4. The experimental results appear somewhat limited, where only a Gaussian-like density and fermion density are investigated. However, I am not familiar with the quantum physics literature and cannot provide perspective on the scope or significance of the results. The fermion results do not match the performance of Ferminet, which might limit the appeal of the proposed method especially since the same architecture is used.
5. While interesting, the formulation of the WSM objective might be somewhat ad-hoc. Is there a more principled way to integrate DMC and VMC beyond an intuitive connection?


**Summary Of The Paper:**

This work proposes a score-based method for learning neural networks to model wavefunctions minimize quantum potentials. The work adapts an objective function from the VMC model that minimizes local energy under current model samples. The main innovation is the proposal to model only the gradients of the log wave function instead of the wave function itself, which is motivated by an insight into the functional form of the local energy. DMC weighting is introduced to overcome optimization difficulties that arise when the wavefunction is not directly available. The proposed method is evaluated on a boson potential and a challenging fermion potential, and experimental results show a good match to physical observations.

**Summary Of The Review:**

I enjoyed reading this paper and appreciate the efforts to develop deep learning methods that are useful in computational quantum physics. I have little knowledge of quantum physics so I can only comment on ML aspects of the paper. Replacing the explicit wave function with a score network is technically relevant and mirrors similar developments of EBM and score models in deep generative modeling. The integration of VMC and DMC is interesting but would be stronger with a more principled formulation if possible. I cannot provide any context on the quality or significance of the experimental results. A major experimental weakness is that the work uses a network with explicit wavefunction for the central experiment even though the motivation of the work is to avoid explicitly modeling the wave function. It seems there is currently no suitable architecture for the proposed method. Overall, I recommend the paper, but I am on the fence.

---

> ### Author Response · Authors · 2022-11-18
> **Reponse to Reviewer PhHJ (Part 2)**
>
> **W3.**
> While interesting, the formulation of the WSM objective might be somewhat ad-hoc. Is there a more principled way to integrate DMC and VMC beyond an intuitive connection?
>
> **A.**
> We have revised the derivation for WSM. Please refer to general response 2.
>
> **Clarity 1.**
> The relation between the WSM objective and $V$ could be explained intuitively to help reader understanding. From what I gather, the score model initially points in many random directions, and the weightings gradually guide the score function to develop "local minima" (for vector fields, this would mean regions where the score function "points to") where the potential $V$ is minimized by causing the model to gradually see many samples of states with lower potential energy. Is it correct to say that the potential $V$ affects learning only through the weightings? It would be helpful to compare the role of $V$ in the proposed work and existing works that have an explicit wavefunction.
>
> **A.**
> Yes, it is correct to say that the potential $V$ affects learning only through the weightings. To find ground states, our goal is to minimize the expectation of local energy, which is composed of the kinetic energy and the potential energy. It is not correct to minimise only over $V$. If we do, then all samples will concentrate to local minimums of $V$. But then the distribution will become extremely sharp and kinetic energy, which is related to the second order derivative of wavefuncion, will be infinite. From an optimization point of view, $V$ is the given constraint in the sample space and we try to find a distribution to minimize the expectation of local energy.
> At ground states, kinetic energy should compensate the fluctuations in potential energy and all samples should have the same local energy.
>
> The role of $V$ in existing works with an explicit wavefunction is the same. Because in all cases we optimize the underlying the distribution representing the wavefunction. Though the wavefunction representations are different.
>
>
> **Clarity 2.**
> The exact objective of the method could be stated more clearly. The second-to-last paragraph of page 2 starts with "Our objective is to find the ground state $\psi_0$”, but the method appears to learn the gradient of the ground state and samples from the ground state. What is the exact primary goal? I suppose that samples from the ground state wavefunction are the practical objective. If the goal is simply to draw samples from the minimum-energy ground states, why not optimize $V(x)$ directly?
>
> **A.**
> The primary goal is to estimate observables, such as kinetic energy or density which can be estimated from sample distributions. We add discussion of the primary goal in the revised text in section 3 after Equation (5).
>
> We cannot optimize $V(x)$ directly. Otherwise the kinetic energy will be infinite. Please refer to the response for Clarity 1.
>
> **Summary.**
> The integration of VMC and DMC is interesting but would be stronger with a more principled formulation if possible. (...) A major experimental weakness is that the work uses a network with explicit wavefunction for the central experiment even though the motivation of the work is to avoid explicitly modeling the wave function. It seems there is currently no suitable architecture for the proposed method.
>
> **A.** We refer to the responses for W3 and W1.
>
> [1] Lin, Jeffmin, Gil Goldshlager, and Lin Lin. "Explicitly antisymmetrized neural network layers for variational Monte Carlo simulation." Journal of Computational Physics (2022): 111765.
>
> [2] Pang, Tianyu, Shuicheng Yan, and Min Lin. "$ O (N^ 2) $ Universal Antisymmetry in Fermionic Neural Networks." arXiv preprint arXiv:2205.13205 (2022).

---

> ### Author Response · Authors · 2022-11-18
> **Reponse to Reviewer PhHJ (Part 1)**
>
> Thanks a lot for your review. We reply to you comments below:
>
> **W1.**
> The Ferminet network used for experiments already provides an estimate for $\psi_0$, so the motivation for score-based modeling is not as clear. What is the motivation or benefit of learning the gradient of a scalar-valued net, instead of learning a network that outputs the estimated gradient directly as typically done in score-based modeling? The fermion restrictions are a difficult challenge with limited options available and the proposed method might be useful for future architectures, but at the present it seems roundabout to use score-based modeling for a scalar valued net.
>
> **A.**
> As you correctly pointed out, the motivation of using the gradient of FermiNet is to handle the fermion sign problem. FermiNet uses the well studied Slater determinant to fulfill the antisymmetry constraint. There are other methods to solve the antisymmtic constraints but they all use wavefuncion-based formulation [1, 2]. To the best of our knowledge, currently there is no principled way to add the antisymmetry constraint directly into score. We could possibly solve this problem by getting inspirations from diffusion Monte Carlo, but so far our attempts to make the score antisymmetric was not successful due to the divergence at nodal surface.
>
> To show that our optimization method can go beyond the energy-based modeling, we add an experiments where the score is modeled as the sum of the FermiNet score and a feed-forward network (FFN). In this way the fermion constraint is satisfied by the first part and the second part could further refine the score. We test this architecture on harmonic traps and atomic systems. Although in the atom case there is no obvious improvement by using FFN, this setting validates the ability to optimize fermions with non wavefunction-based score.
>
>
> **W2.**
> The experimental results appear somewhat limited, where only a Gaussian-like density and fermion density are investigated. However, I am not familiar with the quantum physics literature and cannot provide perspective on the scope or significance of the results. The fermion results do not match the performance of Ferminet, which might limit the appeal of the proposed method especially since the same architecture is used.
>
> **A.**
> The experiments are mainly designed to validate the score-based optimization framework. VMC and DMC have been studied for long time. In contrast, our proposed score-based framework is new. So we think it is important to first show that it can successfully find ground states on systems where exact ground states are known. This is the motivation for harmonic trap experiments. The bosons in harmonic trap indeed have very simple ground states, however, it still requires a correct loss function to get there. In the revised paper, we also include the results for fermions, which is more complicated than bosons (section 4.2).
>
> On the other hand, there are not known analytical solutions for the more challenging atomic systems. It is true that the atom results do not match the performance of Ferminet. We think this is partially due to the difficulty in optimization. In fact, due to using determinants in network, FermiNet requires specially designed second-order optimizer (KFAC) which estimates nature gradients w.r.t. log likelihood outputs. This may not be suitable for our score-based model. More adapted network architectures and optimization methods could be studied in future research.

---

### Author Response · Authors · 2022-11-18
**General Response**

We make the following changes in response to reviewer comments.
1. To show that we can go beyond the FermiNet score in the fermion case, we design a new score network where we combine the FermiNet score with a feedforward network. We test this network with fermions in QHO (section 4.2) and with atoms (section 4.3).
2. We make attempts to derive a rigorous proof for the proposed WSM loss but with no success. Nevertheless, we prove that WSM is unbiased and make more detailed connection from the imaginary time evolution (section 3.3).
3. We add proof for the approximated MH rejection (appendix C).
4. We add more discussions and improve figure quality. Some sections are re-organized to accommodate above changes.

**Edit:**
Additional changes that we would like to include in future versions:
- More clarification about motivations for designing new loss functions instead of using sample mean of local energies (see [discussions](https://openreview.net/forum?id=rMQ1Wme3S0c&noteId=hwfwyzYWVx) with reviewer **afAv**).
- Analysis of additional results and failure cases for Fermions with pure score functions (see [discussions](https://openreview.net/forum?id=rMQ1Wme3S0c&noteId=1iiJRqDE5g4) with reviewer **FLe5**)

---

### Author Response · Authors · 2022-12-11
**Authors' Request for Responses**

We thank the reviewers for taking the time to provide thoughtful comments and helpful suggestions regarding our submission. As the rebuttal period is closing, we ask Reviewer **PhHJ** to please look over our detailed responses and consider updating the review/recommendation based on our clarifying comments, discussions on our novel contributions, and new experiments. We would also like to ask Reviewer **FLe5** to please consider further finalizing the review/recommendation based on other reviewers' feedback and our responses. We are happy to address any remaining questions the reviewers might have.

---

### Decision · Program_Chairs · 2023-01-20

**Decision:**

Reject

**Justification For Why Not Higher Score:**

The paper is clearly borderline and an accept as poster can also be considered.

**Justification For Why Not Lower Score:**

N/A

**Metareview: Summary, Strengths And Weaknesses:**

The paper is located in the research field of neural wavefunctions, i.e. aiming to compute the ground state energy of quantum many-body systems. While recently works which explicitely represent the wavefunction have shown great success, the author propose to follow an implicit approach via a score-based principle. Given the recent success of this area in other domains, this indeed is an exciting direction to investigate. Overall, all reviewers liked the paper and the idea/approach. Two limitations have been identified and discussed in detail among the reviewers:

1) While the authors main aim is to use an "implicit" approach, the final solution they use is basing on the FermiNet architecture (i.e. again using an explicit approach). This is, to some degree, defeating the entire purpose of the model. The authors address this point partially in their feedback -- and also argue that other architectures have now shown better results. Still, it would be interesting to understand why is this the case. And whether this means that explicit models are actually "better" for the considered scenario.

2) The considers scenarios are a bit simplistic -- and the actual benefit, e.g., on the frequently used molecules is not very clear. (E.g. also the runtime/complexity of the method is not better.)

Overall, even after consultation with all reviewers, the paper is borderline.

**Summary Of Ac-Reviewer Meeting:**

see above; all reviewers agreed on the discussed limitations and positive points; it should be noted that the reviewer who provided the highest score indicated uncertainty in some parts of her/his evaluation